# Principled Fine-tuning of LLMs from User-Edits: A Medley of Preference, Supervision, and Reward

**Dipendra Misra⋆∗**
Databricks Mosaic Research

**Aldo Pacchiano⋆**
Boston University

**Ta-Chung Chi**
Databricks Mosaic Research

**Ge Gao**
Google DeepMind

## Abstract

We study how to fine-tune LLMs using user-edit deployment data consisting of a set of context, an agent's response, and user edits. This deployment data is naturally generated by users in applications such as LLMs-based writing assistants and coding agents. The *natural* origin of user edits makes it a desired source for adapting and personalizing of LLMs. In this setup, there emerges a unification of various feedback types namely preferences, supervised labels, and cost that are typically studied separately in the literature. In this paper, we initiate the theoretical investigation of learning from user edits. We first derive bounds for learning algorithms that learn from each of these feedback types. We prove that these algorithms have different trade-offs depending upon the user, data distribution, and model class. We then propose a simple ensembling procedure to jointly learn from these feedback types. On two domains from Gao et al. [21], we show our ensembling procedure outperforms these methods that learn from individual feedback. Further, we show that our proposed procedure can robustly adapt to different user-edit distributions at test time.

## 1 Introduction

Post-training of LLMs has become a critical step in developing large language models (LLMs) in recent times. In particular, high-quality data is a valuable artifact for making LLMs useful for domains of interest. Modern approaches typically employ annotators to label a variety of feedback ranging from full human written gold responses to labeling binary preferences [17, 37]. However, these approaches are inherently expensive. In contrast, some recent works have attempted to use deployment logs generated by natural interactions of users with LLM Agents to improve these models [15, 21, 25]. In particular, for a broad class of LLM applications such as coding and writing assistants, user edits [19, 21] form a natural mode of feedback found in deployment logs. Motivated by this, we study how to fine-tune LLMs to a given task using user-edits in deployment logs.

Consider the application in Figure 1 where an LLM agent is being used as a writing assistant. In any given session, a user can query the agent at any time to solve a task such as generating an email given a short description. If the LLM's response is unsatisfactory, the user typically edits the response to fix the deficiencies. For example, the user may prefer a certain writing style that the base LLM agent may not be tailored to generate. We emphasize that these edits are *naturally generated* by the users making it easy to scale this feedback. Further, since the user has a stake in solving their own problem, we can assume that the edit feedback is of reasonable quality. In contrast, in RLHF applications, feedback such as pairwise preference is collected by hiring annotators to label a pair of

---

∗DM and AP contributed equally and are listed alphabetically. GG contributed before joining DeepMind.

39th Conference on Neural Information Processing Systems (NeurIPS 2025).

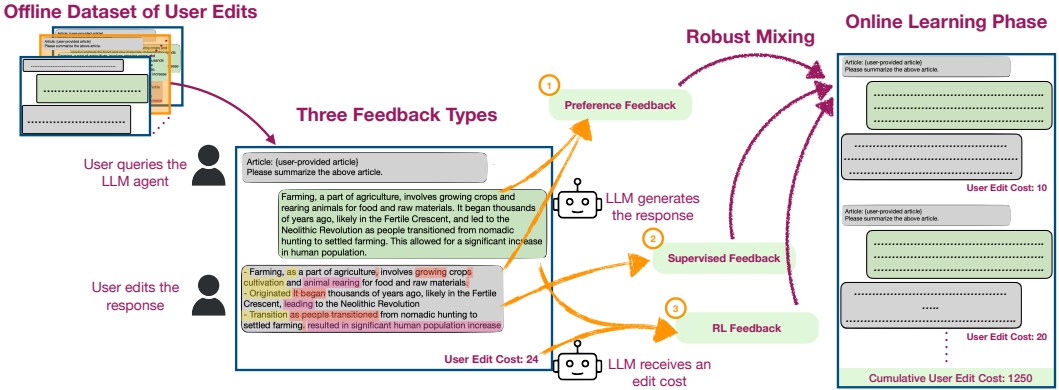

Figure 1: Illustrates our learning from edits setup (Protocol 1). An interesting feature of our setup is that it contains three fundamental types of feedback that can be used to fine-tune policies. We show that using multiple feedback types leads to better policies.

LLM responses and typically requires a quality check. Finally, edit feedback is also convenient to provide since state-of-the-art LLMs are often able to generate a reasonable response that may only need small edits to make it suitable.

There has been significant progress in the development of LLMs over the last few years. However, LLMs are not oracle and cannot apriori adapt to a given user or task preference [47, 52]. A common solution is to prompt these models with the needed preferences and requirement often as a system prompt [32]. However, users in real applications are not prompt-engineers and may not list all their preferences in the prompt, or may even be aware of their preferences. Further, the needs and preferences can be complex, context-dependent, and maybe hard to describe. This necessitates using of user feedback such as edits to improve the agent's response over time.

Our learning protocol shown in Figure 1 consists of an offline learning phase that uses a dataset of deployment logs containing user interactions with the LLM agent. Each data point consists of an original context, along with the agent's response and any user edits. The dataset can be collected from a single user if we are personalizing to a given user's preference, or multiple users if we are learning more general preferences. This dataset is used to adapt the LLM agent. The second phase of our learning consists of an online learning phase during which the agent interacts with a user, or multiple users, to generate a response. We penalize the agent during this phase by the amount of edits, as measured by an edit cost, that the user has to perform to make up for the deficiencies in the agent's response. As in a real-world setting, user(s) during the online learning phase can be misaligned with the user(s) that provided data in the deployment logs. The goal of a learning algorithm is to effectively perform offline and online learning to minimize the cumulative edit cost during the online learning phase.

In machine learning, there is a long line of work on development of optimal algorithms given a set of feedback such as from rewards [51, 48], actions [43, 29], or preferences [17, 37]. As shown in Figure 1, an interesting property of our learning setup is that all three aforementioned feedback types co-occur in this setting. Namely, given the context and user edits, one can simply fine-tune the LLM on the user edits. Alternatively, one can use induce a preference data using the user edits as the preferred response over the agent response. Finally, one can also use the edit distance as a reward to perform reinforcement learning. This provides a rich space to develop algorithms. In this work, we initiate a theoretical investigation into learning from user edits. Our purpose is two-fold: one is to provide principled reasoning for our algorithm choices and compare them to other learning settings, and the second is to predict empirical phenomena that can help in deployment. We first provide bounds for three basic algorithms that use the three separate feedback, and demonstrate that these algorithms have different trade-offs. We then propose an ensembling approach that can balance these different trade-offs.

We evaluate these different methods and our ensembling approach on two domains from Gao et al., [21]. These domains evaluate the ability of an LLM agent to personalize to a simulated user's latent preference based purely on user edit feedback. We show that across a range of different settings, the ensembling approach perform better than algorithms that use a single feedback type.

## 2 Preliminary

**Notation.** We use calligraphic letters such as $\mathcal{U}$ to denote sets. We define $\Delta(\mathcal{U})$ as the set of all distributions over a countable set $\mathcal{U}$. For any $N \in \mathbb{N}$, we define the set $[N] = \{1, 2, \cdots, N\}$. We define the set of all texts as $\mathcal{Y}$ and the set of all contexts $\mathcal{X}$. The context can include text along with other multimodal data such as images or videos. We define the set of LLM policies as $\Pi = \{\pi : \mathcal{X} \to \Delta(\mathcal{Y})\}$. Given a context $x \in \mathcal{X}$, and a policy $\pi \in \Pi$, we can sample a response $y \sim \pi(\cdot \mid x)$. We assume that LLMs can accept multimodal input if the context contains multimodal data.

**Learning from user-edits.** In our setup (Protocol 1), the agent is repeatedly queried by a user to generate a response $y$ given a context $x$. For example, in the writing assistant application, the assistant agent may be triggered to generate an email draft given a short prompt. The context $x$ will include this prompt along with the content of the existing document and application metadata such as environment variables and the agent's calendar information. The agent will then generate a response $y$, which is the email draft. If the user finds the response unsuitable, they may edit it to $y'$. If they decide not to edit, then we have $y' = y$. We will always state there is a user edit, and if $y = y'$ then we will call this an *empty edit*. We assume the user edits can be modeled with an *unknown distribution* $q : \mathcal{X} \times \mathcal{Y} \to \Delta(\mathcal{Y})$ such that given a context $x$ and response $y$, the user generates the edit $y' \sim q(\cdot \mid x, y)$.

We emphasize that the user performs the edits as they need the result for their own needs. This contrasts with the more commonly used preference data in the RLHF literature which is collected by asking annotators to label a pair of responses. As this is not a natural task, the annotators are often compensated. In contrast, *user edits provide freely generated deployment data for performing never-ending fine-tuning of LLMs*.

The necessity for user edits is due to deficiencies in the agent's response $y$. Further, the more edits the user has to make, the more deficient is the agent's response. We can measure this deficiency with an edit cost $c = \Delta_{\text{edit}}(y, y')$ where $\Delta_{\text{edit}} : \mathcal{Y}^2 \to [0, c_{\max}]$ is a suitable edit distance metric. The edit cost $c$ is higher if the user performs more edits and 0 only if edits are empty ($y = y'$). Following Gao et al., [21], we use Levenhstein edit distance over sub-tokens for defining $\Delta_{\text{edit}}$ in our experiments. However, our analysis and algorithm development are independent of the choice of $\Delta_{\text{edit}}$. We are now ready to state our formal setup.

**Formal Learning Setup.** Protocol 1 states our formal setup. We assume access to a dataset of $n$ user edits $\mathcal{D} = \{(x_i, y_i, y'_i\}_{i=1}^n$ and a policy class $\Pi$. For every $i \in [n]$, we assume $x_i \sim \rho(\cdot)$, $y_i \sim \pi_{\text{ref}}(\cdot \mid x_i)$, and $y'_i \sim q(\cdot \mid x_i, y_i)$, where $\rho$ is the context distribution and $\pi_{\text{ref}}$ is a *reference policy* used to generate the agent response. The reference policy can be a pre-trained LLM that is used to warmstart the problem. This is a typical deployment dataset that is encountered in writing and coding assistant applications. This data distribution can be used to perform fine-tune policies in the *offline learning phase* (line 1).

After this, the agent is evaluated over a series of $T$ episodes during the *online learning phase*. In each interaction, the world (which includes the user) will provide a context $x_t \sim \rho$ (line 3). The agent can generate a response $y_t$ for this context (line 6). Finally, the user generates an edit $y'_t$ which leads to an edit cost of $c_t$ to the agent (line 5-6). The goal of the agent is to minimize the total edits performed during these $T$ episodes (line 9). We allow the agent to perform *online learning* during these $T$ episodes. However, $T$ may not be large enough to perform a full fine-tuning. Consequently, any learning algorithm should try to effectively use the edited dataset $\mathcal{D}$.

Our setup can be applied to both settings where edits are performed by an arbitrary set of people or when they are performed by a single person or a particular group. In the first case, we need to learn general preferences, whereas the latter is a *personalization setting* since we need to learn the preferences of a particular person or group.

We define a few useful concepts to enable us to state our formal objective. We first define the expected cost of a response $y$ in context $x$ as $c(x, y) = \mathbb{E}_{y' \sim q(\cdot \mid x, y)} [\Delta_{\text{edit}}(y, y')]$. Given a policy $\pi \in \Pi$, we then define its expected cost as $J(\pi) = \mathbb{E}_{x \sim \rho, y \sim \pi(\cdot \mid x)} [c(x, y)]$. We also define a $\beta$-KL regularized objective as:

$$J_\beta(\pi) = \mathbb{E}_{x \sim \rho, y \sim \pi(\cdot \mid x)} \left[ c(x, y) + \beta \log \frac{\pi(y \mid x)}{\pi_{\text{ref}}(y \mid x)} \right]. \tag{1}$$

**Protocol 1** Finetuning LLMs from User Edits. An algorithm needs to implement the lines in brown.

---

**Require:** Given a dataset $\mathcal{D} = \{(x_i, y_i, y_i')\}_{i=1}^n$ of user edits. Also, given a policy class $\Pi$.
 1: Initialize agent using $\mathcal{D}$                                      // Offline learning phase
 2: **for** $t = 1, 2, 3, \cdots, T$ **do**
 3:     World presents context $x_t$
 4:     Agent generates a response $y_t$                    // Online learning and evaluation phase
 5:     User edits the response to $y_t'$
 6:     Agent receives a cost for edits $c_t = \Delta_{\text{edit}}(y_t, y_t')$
 7: Return $\sum_{t=1}^T c_t$

---

It is well-known that optimal solution for objective in Equation 1 is given by $\pi_\star^\beta(y|x) = \frac{\pi_{\text{ref}}(y'|x)}{Z(x)} \exp\left(\frac{-c(x,y)}{\beta}\right) \pi_{\text{ref}}(y|x)$ where $Z(x) = \sum_{y' \in \mathcal{Y}} \exp\left(\frac{-c(x,y')}{\beta}\right)$ [40]. We assume $\pi_\star^\beta \in \Pi$ for the given $\beta$ and following RLHF literature [37, 40], we compete with this KL-regularized optimal policy. Formally, we define the sub-optimality (SubOpt) of a policy $\pi$ and the regret ($\text{Reg}_T$) of an agent during the online learning phase as:

$$\texttt{SubOpt}(\pi) = J_\beta(\pi) - J_\beta(\pi_\star^\beta), \quad \texttt{Reg}_T = \sum_{t=1}^T \texttt{SubOpt}(\pi_t). \tag{2}$$

where $\pi_t$ is the policy played by the agent in the $t^{th}$ episode. We also define $D(\pi, \pi') = \mathbb{E}_{x \sim \rho}\left[\|\pi(\cdot \mid x) - \pi'(\cdot \mid x)\|_{\text{TV}}\right]$ as the expected total-variation distance between two policies $\pi$ and $\pi'$. Our goal is to find an interactive learning algorithm that learns from the offline dataset $\mathcal{D}$ and minimizes the regret $\text{Reg}_T$ during the online learning phase with high probability.

## 3  Principled Learning from User-Edits

There are two stages of learning in Protocol 1 offline and online. We discuss learning in these two stages below where one has access to a moderately large dataset of user edits and a much more limited number of online learning episodes.

### 3.1  Offline Learning from User-Edits

Our learning protocol (Protocol 1) contains multiple feedback sources that are typically studied separately in the ML literature. We discuss learning algorithms for each feedback source below.

**A. Learning from User Edit Supervision.** User edits $y'$ provide a reasonable sample of the desired behavior. Therefore, one can directly learn a policy $\hat{\pi}_{\text{SUP}}$ by fine-tuning on the user-edits:

$$\hat{\pi}_{\text{SUP}} = \arg\max_{\pi \in \Pi} \ell_{\text{SUP}}(\pi), \quad \text{where} \quad \ell_{\text{SUP}}(\pi) = \frac{1}{n} \sum_{i=1}^n \log \pi(y_i' \mid x_i). \tag{3}$$

**B. Learning from Preferences.** It is possible that users only partially improve the agent's response $y$ when editing it to $y'$ and that the optimal response is still far from $y'$. In this case, fine-tuning on $y'$ can lead to sub-optimal behavior. In contrast, one can use the observation that as users edited $y$ to $y'$, this implies that $y'$ is preferred over $y$ and we can use this to perform learning from preferences. We can use any preference-learning approach with dataset $\{(x_i, y_i, y_i')\}_{i=1}^n$ where $y_i' \succeq y_i$. For example, we can perform Direct Preference Optimization (DPO) [40] to learn policy $\hat{\pi}_{\text{PREF}}$:

$$\hat{\pi}_{\text{PREF}} = \arg\max_{\pi \in \Pi} \ell_{\text{PREF}}(\pi), \quad \text{where} \tag{4}$$

$$\ell_{\text{PREF}}(\pi) = \frac{1}{n} \sum_{i=1}^n \log \sigma\left(\beta \log \frac{\pi(y_i' \mid x_i)}{\pi_{\text{ref}}(y_i' \mid x_i)} - \beta \log \frac{\pi(y_i \mid x_i)}{\pi_{\text{ref}}(y_i \mid x_i)}\right).$$

We emphasize an important difference between the distribution over preferences in Equation 4 and the typical RLHF literature [17, 6, 37], where the preference pair $(y, y')$ is typically generated by independently sampling from the reference policy. In contrast, in our setting only one response is generated from the reference policy whereas the other is generated by user edits. This distributional change impacts both the theoretical analysis and empirical performance.

---

**Algorithm 1** `LateEnsemble` of Policies using Upper Confidence Bound.

---

**Require:** Given a dataset $\mathcal{D} = \{(x_i, y_i, y_i')\}_{i=1}^n$ of user edits. Also, given a policy class $\Pi$.
1: Create a list of policies $\Psi = [\hat{\pi}_{\text{SUP}}, \hat{\pi}_{\text{PREF}}, \hat{\pi}_{\text{RL}}, \hat{\pi}_{\text{EF}}]$ by fine-tuning on $\mathcal{D}$      // Offline learning

2: Define total cost $C(\pi) = 0$ and count $N(\pi) = 0$ for each policy $\pi \in \Psi$
3: **for** $t = 1, 2, 3, \cdots, T$ **do**
4:      World presents context $x_t$
5:      If $t \leqslant |\Psi|$, then $\pi_t = \Psi[t]$ else $\pi_t = \arg\min_{\pi \in \Psi} \left\{ \frac{C(\pi)}{N(\pi)} - \alpha\sqrt{\frac{\log(t)}{N(\pi)}} \right\}$
6:      $y_t \sim \pi_t(\cdot \mid x_t)$      // Online learning
7:      User edits the response to $y_t'$
8:      Agent receives a cost for edits $c_t = \Delta_{\text{edit}}(y_t, y_t')$
9:      Update cost and counts: $C(\pi_t) = C(\pi_t) + c_t$; $N(\pi_t) = N(\pi_t) + 1$.

---

**C. Learning from Cost.** We observe a cost $c_i = \Delta_{\text{edit}}(y_i, y_i')$ for every datapoint which is a sampled from our cost function $c(x, y) = \mathbb{E}_{y' \sim q(\cdot|x,y)} \left[ \Delta_{\text{edit}}(y, y') \right]$. We can, therefore, use this to train a cost model which can be used to perform reinforcement learning. Formally, given a cost model family $\mathcal{F}$ we first learn a cost model via square-loss regression:

$$\hat{f} = \arg\min_{f \in \mathcal{F}} \frac{1}{n} \sum_{i=1}^n \left( f(x_i, y_i) - c_i \right)^2 \tag{5}$$

A direct empirical approach will be to use this cost function and perform RL on an empirical distribution over the contexts in $\mathcal{D}$. However, $\hat{f}$ need not be well-calibrated for all responses. Motivated by this consideration, we define a pessimistic cost function, $\bar{f}(x, y) = \max_{f \in \tilde{\mathcal{F}}} f(x, y)$ where the set of functions $\tilde{\mathcal{F}} = \left\{ f \in \mathcal{F} \text{ s.t. } \sum_{i=1}^n \left( f(x, y) - \hat{f}(x, y) \right) \leqslant \gamma(\mathcal{F}, \delta) \right\}$ for $\gamma(\mathcal{F}, \delta) = \mathcal{O}\left( \log\left( \frac{|\mathcal{F}|}{\delta} \right) \right)$ is a confidence set of cost functions that agree with the historical data. We then train the policy $\hat{\pi}_{\text{RL}}$ to optimize this cost under KL-constraints on our input contexts.

$$\hat{\pi}_{\text{RL}} = \arg\min_{\pi \in \Pi} \mathbb{E}_{i \sim \text{Unf}([n]), y \sim \pi(\cdot|x_i)} \left[ \bar{f}(x_i, y) + \beta \log \frac{\pi(y \mid x_i)}{\pi_{\text{ref}}(y \mid x_i)} \right] \tag{6}$$

While computing the $\bar{f}$ is computationally impractical in the general setting, we nevertheless use this pessimistic RL algorithm in Equation 6 for our theoretical analysis.

**Early-Ensembling of Losses.** As we will demonstrate later, the aforementioned three approaches have different trade-offs. Therefore, it might be advantageous to learn a robust policy by optimizing jointly over their losses. We call such an approach an *early ensembling*. For example, it is quite common in RLHF literature to add the supervised learning loss to the preference learning loss [38]:

$$\hat{\pi}_{\text{EF}} = \arg\min_{\pi \in \Pi} \left( \ell_{\text{PREF}}(\pi) + \lambda \ell_{\text{SUP}}(\pi) \right). \tag{7}$$

We later empirically investigate the form of early ensembling described in Equation 7.

### 3.2 Adaptation During the Online Learning Case

During the online learning phase, we encounter a small number of episodes where the main goal is to evaluate the agent. In practice, this will be the deployment phase where the agent interacts with real users. However, this also provides an opportunity to perform any additional adaptation.

A key challenge of offline approaches is that different methods have different trade-offs. Early-ensembling approaches may not be able to effectively balance between these trade-offs. This is especially true, if the user distribution at test time is different from train time, which can happen in practice. Unfortunately, the small number of online episodes makes it challenging to do any effective fine-tuning. Motivated by these two considerations, we employ a very simple approach of training a set of policies using the different offline learning methods. During the online learning phase, we then run a bandit algorithm to select which of the learned policy to use for generating the response. We use the user-edit cost as input to the bandit algorithm. We call this approach a

*late-ensembling of policies*. Algorithm 1 gives an example using the Upper Confidence Bound (UCB) approach [5, 31, 11], however, other bandit approaches can also be used.

**Pure Online Learning Setting.** In Appendix A, we consider a pure online learning variant of Protocol 1 with a large $T$ and without an offline learning phase, and derive a no-regret algorithm for it.

## 4 Theoretical Analysis of Learning from Edits

In the last decade, significant theoretical understanding has been achieved in reinforcement learning where the feedback is reward, and imitation learning where the feedback is action. Can we achieve the same for learning from edits? We initiate the theoretical understanding of learning from edits to provide a motivation for our algorithm design and predict experimental phenomena.

### 4.1 Theoretical Setup and Assumptions.

We start by stating our modeling assumption for the user distribution in Assumption 1. We assume that the user is more likely to edit a response $y$ to $y'$ instead of the reverse depending upon how much more likely is $y'$ under the optimal policy $\pi^\star$ instead of $y$. This is stated formally in Equation 8 which we call the *balance equation*. This equation can be viewed as serving a similar purpose as the Bradley-Terry distribution in RLHF [10, 17] by providing a smooth characterization of user behavior. We also assume that the user distribution has a non-zero probability of generating the optimal response for any input response $y$. This probability can be small but does not need to scale with the size of the generation space $|\mathcal{Y}|$.

**Assumption 1** (User Distribution)**.** *There exists a (known) $\beta > 0$ such that the user distribution satisfies the* balance equation *below:*

$$\forall x \in \mathcal{X}, y, y' \in \mathcal{Y}, \quad \frac{q(y' \mid x, y)}{q(y \mid x, y')} = \frac{\pi_\star^\beta(y' \mid x)}{\pi_\star^\beta(y \mid x)}. \tag{8}$$

*Further, for any $x \in \mathcal{X}$, the user has at least an $\gamma_{min}(x) > 0$ (possibly user dependent) probability of generating the optimal response $y^\star = \arg\max_{y \in \mathcal{Y}} \pi_\star^\beta(y \mid x)$:*

$$\forall x \in \mathcal{X}, y \in \mathcal{Y}, \quad q(y^\star \mid x, y) \geqslant \gamma_{min}(x) \tag{9}$$

In Appendix A.1, we provide an example where this assumption holds.

**From Balance Equation to Bradley-Terry Distribution.** An important consequence of Assumption 1 is that the preference distribution in Equation 4 satisfies the Bradley-Terry assumption. To see this, we first realize that our preference pairs will contain two responses $(y, y')$ given a context $x$ in precisely two situations: either we generate $y \sim \pi_{\text{ref}}(\cdot \mid x)$ and it is edited to $y' \sim q(\cdot \mid x, y)$, or we generate $y' \sim \pi_{\text{ref}}(\cdot \mid x)$ and it is edited to $y \sim q(\cdot \mid x, y)$. The probability that we prefer $y'$ over $y$ $(y' \succeq y)$ is then given by the probability of the first situation normalized by the joint probability, i.e.,

$$\mathbb{P}(y' \succeq y \mid x, y, y') = \frac{\pi_{\text{ref}}(y \mid x) q(y' \mid x, y)}{\pi_{\text{ref}}(y \mid x) q(y' \mid x, y) + \pi_{\text{ref}}(y' \mid x) q(y \mid x, y')},$$

$$= \frac{\pi_{\text{ref}}(y \mid x) \pi_\star^\beta(y' \mid x)}{\pi_{\text{ref}}(y \mid x) \pi_\star^\beta(y' \mid x) + \pi_{\text{ref}}(y' \mid x) \pi_\star^\beta(y \mid x)}, \quad \text{(from Equation 8)}$$

$$= \sigma\left(c(x, y) - c(x, y')\right), \quad \text{(using } \pi_\beta^\star(\tilde{y} \mid x) = \frac{\pi_{\text{ref}}(y|x)}{Z(x)} \exp(-\frac{c(x,y)}{\beta})\text{)}.$$

This justifies using DPOs even though the joint distribution over the preference pairs $(y, y')$ is different than IID sampling from $\pi_{\text{ref}}$ which is typically studied in the RLHF literature.

As we are working with function approximations, we assume policy realizability, i.e., that our function class is expressive enough to contain the desired policies. This is a standard assumption in theoretical analysis of interactive learning algorithms [28, 54].

**Assumption 2** (Realizability)**.** *We assume that $\pi_\star^\beta \in \Pi$ and $q \circ \pi \in \Pi$ where $q \circ \pi$ is the policy defined as $q \circ \pi(y|x) = \sum_{y' \in \mathcal{Y}} q(y|x, y') \pi(y'|x)$.*

Our first result establishes Assumption 1 implies the edits induce a contraction property on the user distribution,

**Lemma 1.** *For any $\pi \in \Pi$, the user distribution satisfies the following contraction property:*

$$\forall x \in \mathcal{X}, \qquad \left\| q \circ \pi(Y \mid x) - \pi_\star^\beta(Y \mid x) \right\|_{TV} \leqslant (1 - \gamma_{min}(x)) \left\| \pi(Y \mid x) - \pi_\star^\beta(Y \mid x) \right\|_{TV}$$

The proof of Lemma 1 can be found in A. The contraction property implied by Lemma 1 indicates that an iterative application of the user distribution approaches a fixed point.

Finally, we make one final assumption for stating results for DPO. We assume that the data distribution satisfies the following concentrability assumption. Similar to realizability assumptions, concentrability assumptions are typical in RL literature (see for example [28, 54]).

**Assumption 3.** *[Preference Concentrability] For any $\pi \in \Pi$ we assume:*

$$\frac{\mathbb{E}_{(x,\tilde{y},\tilde{y}') \sim Q_\pi} \left[ \left| \beta \log \frac{\pi(\tilde{y}'|x)}{\pi_\star^\beta(\tilde{y}'|x)} - \beta \log \frac{\pi(\tilde{y}|x)}{\pi_\star^\beta(\tilde{y}|x)} \right| \right]}{\mathbb{E}_{(x,\tilde{y},\tilde{y}') \sim Q_{\pi_{\text{ref}}}} \left[ \left| \beta \log \frac{\pi(\tilde{y}'|x)}{\pi_\star^\beta(\tilde{y}'|x)} - \beta \log \frac{\pi(\tilde{y}|x)}{\pi_\star^\beta(\tilde{y}|x)} \right| \right]} \leqslant C_{\text{PREF}},$$

*where $Q_\pi(x, \tilde{y}, \tilde{y}') = \frac{1}{2}\rho(x)\left(\pi(\tilde{y}' \mid x)\pi^\star(\tilde{y} \mid x) + \pi^\star(\tilde{y}' \mid x)\pi(\tilde{y} \mid x)\right)$.*

This assumption states that our data distribution has enough coverage to allow us to evaluate different policies accurately. In practice, we can expect $C_{\text{PREF}}$ to be small when $\Pi$ is a small constrained class around $\pi_{\text{ref}}$. Alternatively, we can consider pessimistic-versions of preference learning methods [27] which avoids paying for covering all policies.

## 4.2 Theoretical Results.

We state the results for our different algorithms below and defer their full proof and analysis to Appendix A. Our first result is a suboptimality bound for the SFT algorithm.

**Theorem 1** (SFT Result). *Let $\epsilon > 0$ be a sub-optimality target. Under Assumption 1 and Assumption 2 to achieve $\texttt{SubOpt}(\hat{\pi}_{SUP}) \leqslant \epsilon + \mathcal{O}\left(\min\left(\eta_{\max} \cdot D(\pi_{\text{ref}}, \pi_\star^\beta), \bar{\eta}_{\max} \cdot D^{1/2}(\pi_{\text{ref}}, \pi_\star^\beta)\right)\right)$ with probability at least $1 - \delta$ it is enough to set $n \geqslant \Omega\left(\frac{\log(|\Pi|/\delta)}{\epsilon^2}\right)$ where $\eta_{\max} = 1 - \min_x \gamma_{min}(x)$ and $\bar{\eta}_{\max} = \sqrt{\mathbb{E}_{x \sim \rho}\left[(1 - \gamma_{min}(x))^2\right]}$ and $D(\pi_{\text{ref}}, \pi_\star^\beta) = \mathbb{E}_{x \sim \rho}\left[\left\|\pi_{\text{ref}}(Y \mid x) - \pi_\star^\beta(Y \mid x)\right\|_{TV}\right]$.*

We state the suboptimality bound for the offline DPO algorithm that treats the edit data set $D = \{(x_i, y_i, y_i')\}_{i=1}^n$ as a preference data set in Theorem 2.

**Theorem 2** (DPO Result). *Let $\epsilon > 0$ be a sub-optimality target. Under Assumption 1, Assumption 2, and Assumption 3 to achieve $\texttt{SubOpt}(\hat{\pi}_{PREF}) \leqslant \epsilon$ with probability at least $1 - \delta$ it is sufficient to set $n \geqslant \Omega\left(\frac{C_{\text{PREF}} \cdot \log(|\Pi|/\delta)}{\beta^2(\sigma'(-V_{\max}))^2\epsilon^2}\right)$ where $V_{max} > 0$ satisfies $\max_{\pi \in \Pi} \max_{(x,y) \in \mathcal{X} \times \mathcal{Y}} \beta \log \frac{\pi(y'|x)}{\pi_{\text{ref}}(y'|x)} \leqslant \frac{V_{max}}{2}$.*

The proof of Theorem 2 is inspired by the techniques of works such as [44, 54]. Our final result characterizes the sample complexity of learning from costs.

**Theorem 3** (RL Result). *Let $\epsilon > 0$ be a sub-optimality target. Under Assumption 2 the policy $\hat{\pi}_{RL}$ satisfies $\texttt{SubOpt}(\hat{\pi}_{RL}) \leqslant \epsilon$ with probability at least $1 - \delta$ when $n \geqslant \Omega\left(\frac{\log(|\Pi|/\delta)}{\epsilon^2} + \frac{\bar{C}_\star^2 \cdot d_{\text{Elud}}(\mathcal{F})\log(|\mathcal{F}|/\delta)}{\epsilon^2}\right)$ where $\bar{C}_\star^2 = \mathbb{E}_{\pi_{\text{ref}}}\left[\left(\frac{\pi_\star^\beta(y|x)}{\pi_{\text{ref}}(y|x)}\right)^2\right]$ is the policy concentrability coefficient of $\pi_{\text{ref}}$ and $d_{\text{Elud}}(\mathcal{F})$ is the Eluder dimension -a measure of statistical complexity- of the cost function class $\mathcal{F}$ (see [45]).*

**Discussion of Trade-Offs.** Theorems 1, 2, and 3 provide some guidance on the cases where using the edit data as a preference dataset is better or worse than doing imitation learning on the edits or learning the cost function and doing RL. When the preference coverage coefficient $C_{\text{PREF}}$ is small, the sample complexity of DPO can be much lower than that of SFT and RL. When the weighted approximation error $\min\left\{\eta_{\max}D(\pi_{\text{ref}}, \pi_\star^\beta), \bar{\eta}_{\max}D^{1/2}(\pi_{\text{ref}}, \pi_\star^\beta)\right\}$ is smaller than the target error $\epsilon$, SFT is the preferred method. Finally, when the approximation error and preference concentrability are large, but policy concentrability is small and the cost function is simple—for instance, a low-dimensional linear function—learning the cost function and then optimizing it is the most sample-efficient approach.

# 5 Experimental Results and Discussion

We empirically evaluate our theoretical findings in this section to see if they apply in complex settings similar to what is encountered in practice.

**Task Setup.** We evaluate on two tasks: email writing and summarization, from Gao et al., [21]. For each task, there exists a dataset of articles from 4 domains. Corresponding to each task and domain, there is a list of latent user preferences described as a *preference string*, for example, "*structured, straight to the points, respectful, professional greeting and closing*". The setup follows Protocol 1. In each round, the agent is given a context describing a task along with a given article. For example, summarize a given article. The *agent does not have access to the latent preference string* but must generate a response that satisfies this preference. An LLM user generates an edit given the response, context, and latent preference string corresponding to the domain of the article in the context. The use an LLM-based user allows for reproducible experiments which facilitates rapid advancement in algorithm development.[2] Finally, we compute edit distance using Levenshtein distance normalized by the number of tokens in the agent response. We use the NLTK word tokenizer to compute the edit distance. We use Qwen 3 32B instruct model as our user but remove think tokens before using the response.

The key challenge of this task is that latent preferences are context-dependent as articles from different domains have different preferences. Further, even for a single domain, there are multiple preferences in a preference string. For example, the aforementioned preference string contains multiple individual preferences such as "*second person narrative*" and "*show emotions*". These two challenges occur in real-world applications. For example, a person may prefer to write informal emails to their friends but write formal reviews for their office reports. Note that the context does not state which domain the article in the given context comes from. An LLM agent must implicitly infer the domain from the context, learn the appropriate preference for it, and then use it to generate an appropriate response.

**Strong and Weak User.** We extend the original setup of Gao et al., [21] to consider two types of users: strong users and weak users. A *strong user* generates edits based on every individual preference in the article's latent preference string. In contrast, a *weak user* samples a subset of preferences in the article's preference string and uses them to generate the edits. For example, in any given interaction a weak user may sample two preferences "*second person narrative*" and "*brief*" and perform edits to satisfy these while ignoring the other preferences in the preference string. The weak user models user who may only prefer to perform small edits at a time. Conceptually, both the weak and strong users have the same optimal behavior. This is because the optimal response that satisfies all the preference for the strong user, also satisfies any subset of preference that the weak user can sample. However, while strong user perform edits that rapidly converges to the optimal policy (higher $\gamma_{\min}$), the weak user's edit will slowly take the response towards the optimal behavior (smaller $\gamma_{\min}$).

**Offline Dataset.** We collect an offline interaction dataset between the agent model and the user. This dataset is supposed to represent deployment logs found in typical LLM agent applications for writing and coding assistants. We use Llama 3.1 8b Instruct model as our agent model. We perform generate responses by doing greedy decoding. We collect a dataset of 20,000 examples for the summarization task and 10,000 examples for the email writing task. We collect these datasets separately with both strong and weak users. This gives us 4 separate offline datasets.

**Online Learning Phase.** We evaluate the model for $T = 200$ examples for summarization and email writing. We always use the strong user during test time regardless of which user was used to generate the offline dataset. We run each experiment with 3 different seeds.

**Methods and Implementation.** We consider the following approaches: (i) `Base` which generates from the base agent model, (ii) `SFT` which performs SFT on the training data, (iii) `DPO` which runs the DPO algorithm on the training data, (iv) `EarlyEnsemble` which performs early ensembling of `DPO` and `SFT` losses (Equation 7), and (v) `LateEnsemble` which performs late ensembling (Algorithm 1) using policies trained by methods (ii)-(iv). All generations are performed greedily and with a max number of generation tokens of 1000. We do not evaluate Equation 6 give challenges in implementing it.

We sweep over various hyperparameters including learning rate and epochs for `SFT` and `DPO`, as well as $\beta$ for `DPO`, and $\beta, \lambda$ for `EarlyEnsemble`. We pick the best hyperparameters by maximizing log-loss on a held-out validation set of 200 user-edits. We found in early studies that despite sweeping over the hyperparameters, both `DPO` and `EarlyEnsemble` learn policies that tend to repeat text

---

[2]Gao et al., [21] took steps to validate their LLM user. Please see their paper for details.

| Method | Summarization | | Email Writing | | |
|---|---|---|---|---|---|
| | Strong User | Weak User | Strong User | Weak User | Max SubOpt |
| Base | $0.9455_{+0.01}$ | $0.9445_{+0.02}$ | $0.5108_{+0.03}$ | $0.4923_{+0.01}$ | $0.7364_{+0.10}$ |
| SFT | $0.5377_{+0.02}$ | $0.9304_{+0.19}$ | $0.4159_{+0.05}$ | $0.4539_{+0.03}$ | $0.5772_{+0.19}$ |
| DPO | $1.0790_{+0.06}$ | $0.8267_{+0.06}$ | $\mathbf{0.3365}_{+0.00}$ | $\mathbf{0.3368}_{+0.01}$ | $0.8698_{+0.11}$ |
| EarlyEnsemble | $\mathbf{0.2092}_{+0.09}$ | $\mathbf{0.3586}_{+0.01}$ | $0.3438_{+0.06}$ | $0.4864_{+0.01}$ | $0.1612_{+0.01}$ |
| LateEnsemble | $0.2768_{+0.13}$ | $0.4403_{+0.03}$ | $0.4202_{+0.11}$ | $0.3739_{+0.04}$ | $\mathbf{0.1586}_{+0.04}$ |

Table 1: Results on the summarization and email writing domain. We calculate the mean total edit distance $\frac{1}{T}\sum_{t=1}^{T} c_t$ during the online learning phase. We report average performance across 3 seeds.

| Method | Summarization | | Email Writing | | |
|---|---|---|---|---|---|
| | Strong User | Weak User | Strong User | Weak User | Max. SubOpt |
| Base | $0.8498_{+0.01}$ | $0.8558_{+0.01}$ | $0.4241_{+0.01}$ | $0.4239_{+0.01}$ | $0.6654_{+0.01}$ |
| SFT | $0.4890_{+0.01}$ | $0.7698_{+0.01}$ | $0.3377_{+0.02}$ | $0.3866_{+0.01}$ | $0.5121_{+0.02}$ |
| DPO | $0.9650_{+0.03}$ | $0.7800_{+0.02}$ | $0.2955_{+0.01}$ | $\mathbf{0.3047}_{+0.02}$ | $0.7805_{+0.02}$ |
| EarlyEnsemble | $\mathbf{0.1845}_{+0.02}$ | $\mathbf{0.2577}_{+0.02}$ | $\mathbf{0.2406}_{+0.03}$ | $0.4001_{+0.03}$ | $0.0955_{+0.03}$ |
| LateEnsemble | $0.2509_{+0.04}$ | $0.3403_{+0.01}$ | $0.3123_{+0.01}$ | $0.3428_{+0.03}$ | $\mathbf{0.0862}_{+0.02}$ |

Table 2: Transfer learning results with a Llama-3.3-70B-Instruct User at test time.

until their max tokens expire. We, therefore, perform a post-generation trimming strategy where we trim the generations once it starts repeating 200 consecutive characters. For a fair comparison, we apply this post-processing to the output of all methods. An alternative approach could be to use explicit length penalty [39] or more stable preference-learning approaches such as Rebel [22]. See Appendix C for full details on experimental setting.

**Main Results.** We report the main results in Table 1 with the Qwen-3 32B user. We report mean edit cost across rounds ($T$) and seeds. We also report *Max. SubOpt* that measures worst-case performance gap across the four settings between a given approach and the best performance. We see that `Base` method does not perform well showing the need for adaptation. Intuitively, we would expect a strong user to have strong convergence to the optimal policy. In our theory, this would be reflected in the value of $\gamma_{\min}$ in Assumption 1. We see this in our results where SFT is able to learn and performs much better when trained and tested on strong user. Its performance when trained on weak user is weaker consistent with Theorem 1. In contrast, DPO's theoretical analysis does not depend on the choice of $\gamma_{\min}$ and instead only depends on coverage and the validity of balance equation. We see that DPO performs better than SFT when we train on weak user and test on strong user. However, when both users are strong, then there is no clear winner between SFT and DPO. Overall, this illustrates the trade-off predicted by our theory where SFT can take advantage whenever the user-edits converges faster to the optimal policy, however, it is also susceptible when this is not the case whereas DPO is more robust to this factor.

We see that `EarlyEnsemble` can exploit these trade-offs to achieve better performance than both SFT and DPO on the summarization domain, however, it under-performs DPO on the email writing domain. One explanation can be sensitivity to the choice of $\lambda$. While we perform a grid search over this hyperparameter, and use a validation loss over user-edits, it is possible that this selection scheme does not correspond well with test-time performance. Finally, our `LateEnsemble` approach is able to more readily adapt at test time by making suitable trade-offs and achieve the lowest worst case SubOpt value. We emphasize that late ensembling isn't guaranteed to do better in practice. Specifically, if one of the policy in $\Psi$ on line 1 in Algorithm 1, is always better than others, then UCB will achieve lower performance than this policy due to exploring other sub-optimal policies.

**Transfer Learning to Different User LLM.** We investigate how robust these methods are to a different user LLM. This mimics a common real-world setting where the training data may come from one set of users, however, testing maybe done with a different set of users. These two sets of users can still have a common preference because these maybe dependent on organization, platform, or other controlled features such as age, or location. We use Llama-3.3-70B-Instruct model as our user model during the online testing phase while using our previously trained models. Table 2 shows the results. We see that the main trends from Table 1 carry over even when we test with a different user LLM. In particular, our ensemble approach `LateEnsemble` achieves the best SubOpt value.

Table 3: Generations from `EarlyEnsemble` on the summarization task with a strong user.

| Type | (Latent User Preference String) Agent Response |
|---|---|
| Success | (***question answering style!***) **Q: What was the state of the land and people after the calamity?** A: The land was exhausted, the towns were stripped of their fighting population, and only a shadow of a kingdom remained. ··· [truncated] |
| | (***bullet points, parallel structure, brief***) **Updated Mn PBPK model**: Reflects rapid Mn association/dissociation in tissues, with saturation of binding sites limiting accumulation. |
| | - **Key changes**: |
| | - Rapid Mn binding and dissociation rates. |
| | - High-affinity binding with rapid exchange between bound and free forms. |
| | - Tissue-specific binding capacities to account for background Mn levels. |
| | - **Model performance**: Successfully describes Mn kinetics in adult rats exposed to ··· [truncated] |
| Failure | (***inquisitive, skillful foreshadowing, with emojis***) **Q: What is the traditional approach to drug development, and why is it no longer effective?** |
| | A: The traditional approach focuses on a single target protein, disease, and drug. However, this paradigm is now considered untenable due to the promiscuity of drug-like compounds, which can interact with multiple target proteins. |
| | **Q: How do current target-fishing approaches address the promiscuity issue?** |
| | A: Current approaches, such as single-label multi-classification schemes, ··· [truncated] |
| | (***question answering style***) The article compares two ministers, Mr. Mearns and Rev. Mr. Tindall, highlighting their differences in style and delivery. Mr. Mearns is described as steady, patient, and well-spoken, ··· [truncated] |

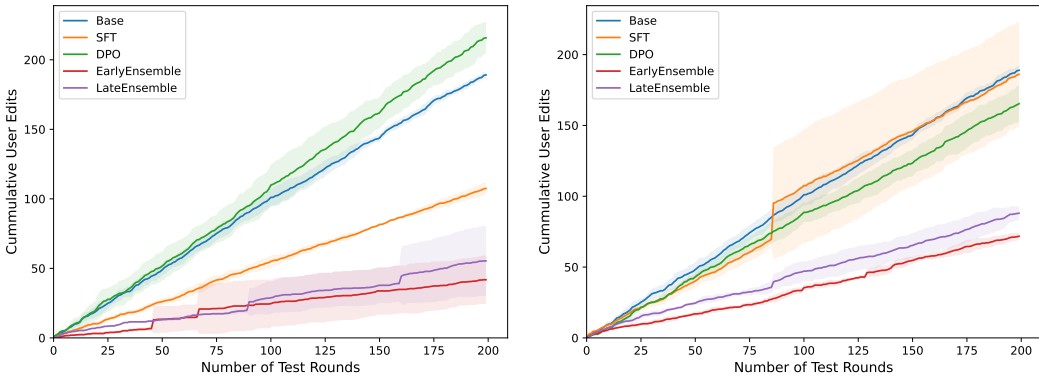

(a) Summarization Domain with Strong User     (b) Summarization Domain with Weak User

Figure 2: Cumulative User Edits at test time corresponding to Table 1 for the different setup. For each round, we show mean and standard deviation across 3 seeds.

**Sampling Generations and Test-time plot**. We provide a few sample generations showing some success and failure cases in Table 3. We also visualize the test-time performance of different methods on the summarization task in Figure 2. As we can see, the `LateEnsemble` approach converges to the best method in both cases as expected. We provide additional details in Appendix C.

# 6 Conclusion and Limitations.

User edits provide a natural feedback for adapting LLMs. In this work, we initiated the theoretical investigation into learning from user edits. We propose a learning setup (Protocol 1) and analyze different algorithms in this setting. We show that none of these algorithms is always desired and propose a simple ensembling approach that empirically works best in two domains. Future works can investigate limitations of our work such as relaxing different assumptions, deriving optimal algorithms, and evaluating these algorithms in a human study.

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

# Appendix

## Table of Contents

## A  Theoretical Analysis with Edit Distance

We provide a table of notations for convenience in Table 4.

| Notation | Description |
|---|---|
| $\Delta(\mathcal{U})$ | Set of all distributions over a countable set $\mathcal{U}$. |
| $[N]$ | Denotes the set $\{1, 2, \cdots, N\}$ for any $N \in \mathbb{N}$. |
| $\mathcal{X}$ | List of contexts which can include text, images, etc. |
| $\mathcal{Y}$ | List of text-based agent response. |
| $\rho \in \Delta(\mathcal{X})$ | Distribution over context at both train time and test time |
| $q : \mathcal{X} \times \mathcal{Y} \to \Delta(\mathcal{Y})$ | User distribution, given a context $x \in \mathcal{X}$ and agent response $y \in \mathcal{Y}$, the user generate an edited response $y' \in \mathcal{Y}$ with probability $q(y' \mid x, y)$. |
| $\Delta_{\text{edit}} : \mathcal{Y}^2 \to [0, c_{\max}]$ | Computes edit distance $\Delta_{\text{edit}}(y, y')$ for editing $y$ to $y'$. We assume $\Delta_{\text{edit}}(y, y) = 0$. |
| $c : \mathcal{X} \times \mathcal{Y} \to [0, c_{\max}]$ | Expected edit cost $c(x, y)$ for response $y$ in context $x$. |
| $\pi : \mathcal{X} \to \Delta(\mathcal{Y})$ | A general notation for a policy $\pi$ that generates a response $y$ given context $x$ with probability $\pi(y \mid x)$. |
| $\Pi$ | A list of policies. |
| $\beta$ | KL-divergence coefficient. It is always a non-negative coefficient. |
| $\pi_\beta^\star$ | KL-regularized optimal policy |
| $\hat{\pi}_{\text{SUP}}$ | Policy trained with supervised fine-tuning (`SFT`). |
| $\hat{\pi}_{\text{PREF}}$ | Policy trained with Direct Preference Optimization (`DPO`). |
| $\hat{\pi}_{\text{RL}}$ | Policy trained with reinforcement learning ($\hat{\pi}_{\text{RL}}$). |
| $\hat{\pi}_{\text{EF}}$ | Policy trained with early ensembling (`EarlyEnsemble`). |

Table 4: Table of important notations and their description.

### A.1  Properties of the Balance Equation

Our setup consists of a distribution $\rho \in \Delta(\mathcal{X})$ over context $x$, a user edit distribution $q : \mathcal{X} \times \mathcal{Y} \to \Delta(\mathcal{Y})$, and an edit distance $\Delta_{\text{edit}} : \mathcal{Y} \times \mathcal{Y} \to [0, c_{\max}]$. We make two assumptions on our setup:

$$\forall x \in \mathcal{X}, y, y' \in \mathcal{Y}, \qquad \frac{q(y' \mid x, y)}{q(y \mid x, y)} = \frac{\pi_\star^\beta(y' \mid x)}{\pi_\star^\beta(y \mid x)},$$

and that for every $x \in \mathcal{X}$, there exists $y^\star$ such that for every $y \in \mathcal{Y}$, we have $q(y^\star \mid x, y) \geqslant \gamma_{\min} = \min_x \gamma_{\min}(x)$. For simplicity we consider a constant $\gamma_{\min}$ function in this section. We now consider examples of this setup.

**Example 1.** We start with a singleton context set $\mathcal{X} = \{x\}$ and $\mathcal{Y} = \{y_1, y_2, \cdots, y_N\}$. We define the user edit distribution below:

$$q(y_N \mid y_i, x) = \gamma_{\min} + \frac{1 - \gamma_{\min}}{N}, \qquad q(y_j \mid x, y_i) = \frac{1 - \gamma_{\min}}{N}, \forall j \neq N$$

We define edit distance as follows $\Delta_{\text{edit}}(y, y') = \delta\{y \neq y'\}$ for any $y$ and $y'$. Using this, we get the expected cost as $c(x, y) = \delta - \delta\frac{1-\gamma_{\min}}{N}$ for every $y \neq y_N$ and $c(x, y_N) = \delta - \delta\left(\gamma_{\min} + \frac{1-\gamma_{\min}}{N}\right)$. Let $\bar{c} = c(x, y_N)$ then for every $y \neq y_N$ we have $c(x, y) = \bar{c} + \delta\gamma_{\min}$. Hence, the preferred response is $y_N$. We also satisfy the constrained that we must have at least $\gamma_{\min}$ going to the optimal response. That leaves only the balance equation. The optimal policy is given by:

$$\pi_\star^\beta(y \mid x) = \frac{1}{Z_\beta(x)} \exp\left(-\frac{c(x, y)}{\beta}\right)$$

For any $y, y' \neq y_N$ we have $q(y' \mid x, y) = q(y \mid x, y') = \frac{1-\gamma_{\min}}{N}$. Further, we also have $\pi_\star^\beta(y \mid x) = \pi_\star^\beta(y' \mid x) = \frac{1}{Z_\beta(x)} \exp\left(-\frac{\bar{c}+\delta\gamma_{\min}}{\beta}\right)$. Therefore, the balance equation is satisfied in this case. When both are $y_N$ then the balance equation is also satisfied. This leaves a single case where one of them is $y_N$ and other is not. Let the other be $y$.

We have:

$$\frac{q(y_N \mid x, y)}{q(y \mid x, y_N)} = \frac{\gamma_{\min} + \frac{1-\gamma_{\min}}{N}}{\frac{1-\gamma_{\min}}{N}} = \frac{1 + N\gamma_{\min} - \gamma_{\min}}{1 - \gamma_{\min}}$$

and

$$\frac{\pi^\star(y_N \mid x)}{\pi^\star(y \mid x)} = \exp\left(\frac{c(x, y) - c(x, y_N)}{\beta}\right) = \exp\left(\frac{\delta\gamma_{\min}}{\beta}\right)$$

Combining them we get:

$$\frac{1 + N\gamma_{\min} - \gamma_{\min}}{1 - \gamma_{\min}} = \exp\left(\frac{\delta\gamma_{\min}}{\beta}\right) \tag{10}$$

It is straightforward to see that for any choice of $N$ and $\gamma_{\min}$, we can define $\beta$ to satisfy this last equation.

### A.1.1 Contraction Property

**Lemma 2** ($\pi^\star$ as steady-state of user distribution)**.** *For any $x \in \mathcal{X}$ we have $q \circ \pi_\star^\beta(Y \mid x) = \pi_\star^\beta(Y \mid x)$.*

*Proof.* Fix $x \in \mathcal{X}$, then for any $y, y' \in \mathcal{Y}$ we have from the balance equation in Assumption 1 the following:

$$\frac{q(y' \mid x, y)}{q(y \mid x, y')} = \frac{\pi_\star^\beta(y' \mid x)}{\pi_\star^\beta(y \mid x)} \Rightarrow q(y' \mid x, y)\pi_\star^\beta(y \mid x) = \pi_\star^\beta(y' \mid x)q(y \mid x, y') \tag{11}$$

Summing over $y$ on both sides gives:

$$\sum_{y \in \mathcal{Y}} q(y' \mid x, y)\pi_\star^\beta(y \mid x) = \pi_\star^\beta(y' \mid x) \sum_{y \in \mathcal{Y}} q(y \mid x, y'),$$

$$= \pi_\star^\beta(y' \mid x).$$

This is equivalent to $q \circ \pi^\star = \pi^\star$. $\qquad\square$

**Lemma 1.** *For any $\pi \in \Pi$, the user distribution satisfies the following contraction property:*

$$\forall x \in \mathcal{X}, \qquad \left\| q \circ \pi(Y \mid x) - \pi_\star^\beta(Y \mid x) \right\|_{TV} \leqslant (1 - \gamma_{min}(x)) \left\| \pi(Y \mid x) - \pi_\star^\beta(Y \mid x) \right\|_{TV}$$

*Proof.* Fix $x \in \mathcal{X}$. We start by using Lemma 2 to express:

$$\left\|q \circ \pi(Y \mid x) - \pi_\star^\beta(Y \mid x)\right\|_{\text{TV}} = \left\|q \circ \pi(Y \mid x) - q \circ \pi_\star^\beta(Y \mid x)\right\|_{\text{TV}},$$

$$= \frac{1}{2} \sum_{y' \in \mathcal{Y}} \left| \sum_{y \in \mathcal{Y}} q(y' \mid x, y)\pi(y \mid x) - \sum_{y \in \mathcal{Y}} q(y' \mid x, y)\pi_\star^\beta(y \mid x) \right|$$

We define a quantity $w(y', y; x)$ as follows:

$$w(y', y; x) = \begin{cases} q(y^\star \mid x, y) - \gamma_{\min}(x), & \text{if } y' = y^\star \\ q(y' \mid x, y), & \text{otherwise} \end{cases}$$

Assumption 1 implies $w(y', y; x) \geqslant 0$ and

$$\sum_{y' \in \mathcal{Y}} w(y', y; x) = w(y^\star, y; x) \tag{12}$$

$$+ \sum_{y' \in \mathcal{Y}, y' \neq y^\star} w(y', y; x) = q(y^\star \mid x, y) - \gamma_{\min}(x) + \sum_{y' \in \mathcal{Y}, y' \neq y^\star} q(y' \mid x, y),$$

$$= 1 - \gamma_{\min}(x).$$

Plugging it, we get:

$$\left\|q \circ \pi(Y \mid x) - \pi_\star^\beta(Y \mid x)\right\|_{\text{TV}}$$

$$= \frac{1}{2} \left| \sum_{y \in \mathcal{Y}} \left\{w(y^\star, y; x) + \gamma_{\min}(x)\right\} \pi(y \mid x) - \sum_{y \in \mathcal{Y}} \left\{w(y^\star, y; x) + \gamma_{\min}(x)\right\} \pi_\star^\beta(y \mid x) \right|$$

$$+ \frac{1}{2} \sum_{y' \in \mathcal{Y}, y' \neq y^\star} \left| \sum_{y \in \mathcal{Y}} w(y', y; x)\pi(y \mid x) - \sum_{y \in \mathcal{Y}} w(y', y; x)\pi_\star^\beta(y \mid x) \right|,$$

$$= \frac{1}{2} \sum_{y' \in \mathcal{Y}} \left| \sum_{y \in \mathcal{Y}} w(y', y; x)\pi(y \mid x) - \sum_{y \in \mathcal{Y}} w(y', y; x)\pi_\star^\beta(y \mid x) \right|$$

We view $w(y', y; x)$ as a $|\mathcal{Y}| \times |\mathcal{Y}|$ matrix $W$ with entries $W_{y'y} = w(y', y; x) \geqslant 0$ satisfying $\sum_{y'} W_{y',y} = 1 - \gamma_{\min}(x)$. We omit $x$ from the $W$ notation as it is fixed. Similarly, we define $|\mathcal{Y}|$-dimensional vectors $\pi$ and $\pi^\star$ with entries $\pi(y \mid x)$ and $\pi_\star^\beta(y \mid x)$. Using these matrix and vector notation we can express:

$$\left\|q \circ \pi(Y \mid x) - \pi^\star(Y \mid x)\right\|_{\text{TV}} = \frac{1}{2} \sum_{y' \in \mathcal{Y}} \left| \sum_{y \in \mathcal{Y}} w(y', y; x)\pi(y \mid x) - \sum_{y \in \mathcal{Y}} w(y', y; x)\pi_\star^\beta(y \mid x) \right|$$

$$\leqslant \frac{1}{2} \sum_{y \in \mathcal{Y}} \sum_{y' \in \mathcal{Y}} w(y', y; x) \cdot \left|\pi(y \mid x) - \pi_\star^\beta(y \mid x)\right|$$

$$= \frac{1}{2} \sum_{y \in \mathcal{Y}} (1 - \gamma_{\min}(x)) \cdot \left|\pi(y \mid x) - \pi_\star^\beta(y \mid x)\right|$$

$$= (1 - \gamma_{\min}(x)) \cdot \left\|\pi(Y|x) - \pi_\star^\beta(Y|x)\right\|_{\text{TV}}$$

where the second step uses the non-negatitivy of $w(y'y; x)$ and triangle inequality and the third step uses Equation 12. $\qquad\square$

## A.2 General Results

Recall that we define optimal policy as a KL-regularized policy:

$$\pi_\star^\beta = \arg\min_\pi J_\beta(\pi), \text{ where} \tag{13}$$

$$J_\beta(\pi) = \mathbb{E}_{x \sim \rho, y \sim \pi} \left[c(x, y) + \beta D_{\text{KL}}(\pi || \pi_{\text{ref}})\right], \tag{14}$$

$\beta > 0$ is a chosen hyperparameter defining the objective. We can establish easily that $\pi^\star$ is given by:

$$\pi_\star^\beta(y \mid x) \propto \pi_{\mathrm{ref}}(y \mid x) e^{-\beta c(x,y)}$$

This is a standard result but we prove it in the next Lemma for completion.

**Lemma 3** (Optimal Policy). *The optimal policy satisfies* $\pi_\star^\beta(y \mid x) = W^\star(x)\pi_{\mathrm{ref}}(y \mid x) e^{-c(x,y)/\beta}$ *for every* $x \in \mathcal{X}, y \in \mathcal{Y}$, *where* $W^\star(x) = \sum_{y' \in \mathcal{Y}} \pi_{\mathrm{ref}}(y' \mid x) e^{-c(x,y')/\beta}$ *is the normalization constant.*

*Proof.* The Lagrangian for the problem is given by:

$$\mathcal{L} = J_\beta(\pi) + \sum_x \mu_x \left( \sum_y \pi(y \mid x) - 1 \right),$$

where $\{\mu_x \mid x \in \mathcal{X}\}$ are our Lagrange multipliers. Taking derivative wrt to $\pi(y \mid x)$ for a fixed value of $x \in \mathcal{X}$ and $y \in \mathcal{Y}$ and setting it to 0 we get:

$$\frac{\partial \mathcal{L}}{\partial \pi(y \mid x)} = \rho(x)c(x,y) + \beta\rho(x)\log \pi(y \mid x) + \beta\rho(x) - \beta\rho(x)\log \pi_{\mathrm{ref}}(y \mid x) + \mu_x = 0$$

Rearranging the terms we get:

$$\log \frac{\pi(y \mid x)}{\pi_{\mathrm{ref}}(y \mid x)} + 1 + \frac{\mu_x}{\beta\rho(x)} = -\frac{c(x,y)}{\beta}$$

let $W^\star(x) = \exp\left(-1 - \frac{\mu_x}{\beta\rho(x)}\right)$, we get:

$$\pi(y \mid x) = W'(x)\pi_{\mathrm{ref}}(y \mid x) e^{-\frac{c(x,y)}{\beta}},$$

the value of $W^\star(x)$ serves as a normalization factor to ensure $\sum_{y \in \mathcal{Y}} \pi(y \mid x) = 1$ and is enforced by taking derivative wrt to $\mu_x$ and setting them to 0. This gives $W^\star(x) = \sum_{y' \in \mathcal{Y}} \pi_{\mathrm{ref}}(y' \mid x) e^{-c(x,y')/\beta}$. $\square$

Recall that we define $\mathtt{SubOpt}(\pi) = J_\beta(\pi_\star^\beta) - J_\beta(\pi)$. In many practical setting, we want to compete with the regularized policy but we only care about the unregularized cost. We capture this by defining the *unregularized sub-optimality*:

$$\mathtt{SubOpt}_0(\pi) = J_0(\pi) - J_0(\pi_\star^\beta) \tag{15}$$

We also define *unregularized regret* as

$$\mathtt{Reg}_T^0 = \sum_{t=1}^T \mathtt{SubOpt}_0(\pi_t), \tag{16}$$

where $\pi_t$ is the policy played by the agent in round $t$. If we play a fixed policy $\pi$ for all $T$ rounds then we have $\mathtt{Reg}_T^0 = \sum_{t=1}^T \mathtt{SubOpt}_0(\pi) = T\mathtt{SubOpt}_0(\pi)$. In these cases, we will use $\mathtt{Reg}_T^0(\pi)$ to denote the regret of this fixed policy.

A useful lemma below relates the total variation between policies to the suboptimality.

**Lemma 4** (TV to Unregularized Suboptimality). *If* $\mathbb{E}_{x \sim \rho} \left[ \left\| \pi(Y \mid x) - \pi_\star^\beta(Y \mid x) \right\|_{TV} \right] \leqslant \epsilon$, *then* $\mathtt{SubOpt}_0(\pi) \leqslant 2\epsilon c_{max}$ *and* $\mathtt{Reg}_T^0(\pi) \leqslant 2\epsilon c_{max}T$.

*Proof.*

$$\mathtt{SubOpt}_0(\pi) = J_0(\pi) - J_0(\pi^\star) = \mathbb{E}_{x \sim \rho} \left[ \sum_{y \in \mathcal{Y}} \pi(y \mid x)c(x,y) - \sum_{y \in \mathcal{Y}} \pi_\star^\beta(y \mid x)c(x,y) \right],$$

$$\leqslant \mathbb{E}_{x \sim \rho} \left[ \sum_{y \in \mathcal{Y}} \left| \pi(y \mid x) - \pi_\star^\beta(y \mid x) \right| c(x,y) \right],$$

$$= 2c_{\max} \mathbb{E}_{x \sim \rho} \left[ \left\| \pi(Y \mid x) - \pi_\star^\beta(Y \mid x) \right\|_{TV} \right] \leqslant 2\epsilon c_{\max}.$$

Finally, $\mathtt{Reg}_T^0(\pi) = T\mathtt{SubOpt}_0(\pi) \leqslant 2\epsilon c_{\max}T$ $\square$

**Bounding Regularized Suboptimality and Regret.** One challenge in bounding regularized suboptimality $\texttt{SubOpt}(\pi) = J_\beta(\pi) - J_\beta(\pi_\star^\beta)$ and regularized regret $\texttt{Reg}_T = \sum_{t=1}^T \texttt{SubOpt}(\pi_t)$ is dealing with KL divergence. E.g., a policy $\pi$ can satisfy $\left\| \pi(Y \mid x) - \pi_\star^\beta(Y \mid x) \right\|_{\text{TV}} \leq \epsilon$ for some $x$ but still have $D_{\text{KL}}(\pi(Y \mid x) \| \pi_{\text{ref}}(Y \mid x)) = \infty$ if $\pi(y \mid x) = \epsilon$, $\pi_\star^\beta(y \mid x) = \pi_{\text{ref}}(y \mid x) = 0$.

However, in RLHF analysis it is often assumed that densities are bounded [44, 54] and this allows us to avoid these cases. In fact, we will later assume this in our proof of DPO. Specifically, we have

**Assumption 4** (Bounded Density Ratios). *We assume there exists $V_{max} > 0$, such that:*

$$\forall \pi \in \Pi, x \in \mathcal{X}, y \in \mathcal{Y}, \qquad \left| \log \frac{\pi(y \mid x)}{\pi_{\text{ref}}(y \mid x)} \right| \leq \frac{V_{max}}{\beta}.$$

Under this assumption we can bound the regularized suboptimality and regret.

**Lemma 5** (TV to Regularized Suboptimality). *If $\mathbb{E}_{x \sim \rho} \left[ \left\| \pi(Y \mid x) - \pi_\star^\beta(Y \mid x) \right\|_{TV} \right] \leq \epsilon$, then under Assumption 4, we have $\texttt{SubOpt}(\pi) \leq 2\epsilon(c_{max} + V_{max})$ and $\texttt{Reg}_T(\pi) \leq 2\epsilon(c_{max} + V_{max})T$.*

*Proof.* We define shorthand $\mathbb{E}_{\pi_1} \left[ \log \frac{\pi_2}{\pi_3} \right] = \mathbb{E}_{x \sim \rho(\cdot), y \sim \pi_1(y|x)} \left[ \log \frac{\pi_2(y|x)}{\pi_3(y|x)} \right]$. Then we have:

$\texttt{SubOpt}(\pi) = J_\beta(\pi) - J_\beta(\pi_\star^\beta)$

$$= J_0(\pi) - J_0(\pi_\star^\beta) + \beta \mathbb{E}_\pi \left[ \log \frac{\pi}{\pi_{\text{ref}}} \right] - \beta \mathbb{E}_{\pi_\star^\beta} \left[ \log \frac{\pi_\star^\beta}{\pi_{\text{ref}}} \right]$$

$$\leq 2\epsilon c_{\max} + \beta \mathbb{E}_\pi \left[ \log \frac{\pi}{\pi_{\text{ref}}} \right] - \beta \mathbb{E}_{\pi_\star^\beta} \left[ \log \frac{\pi_\star^\beta}{\pi_{\text{ref}}} \right]$$

$$= 2\epsilon c_{\max} + \beta \mathbb{E}_\pi \left[ \log \frac{\pi}{\pi_{\text{ref}}} \right] - \beta \mathbb{E}_{\pi_\star^\beta} \left[ \log \frac{\pi}{\pi_{\text{ref}}} \right] + \beta \mathbb{E}_{\pi_\star^\beta} \left[ \log \frac{\pi}{\pi_{\text{ref}}} \right] - \beta \mathbb{E}_{\pi_\star^\beta} \left[ \log \frac{\pi_\star^\beta}{\pi_{\text{ref}}} \right]$$

$$= 2\epsilon c_{\max} + \beta \mathbb{E}_\pi \left[ \log \frac{\pi}{\pi_{\text{ref}}} \right] - \beta \mathbb{E}_{\pi_\star^\beta} \left[ \log \frac{\pi}{\pi_{\text{ref}}} \right] - \beta \mathbb{E}_{\pi_\star^\beta} \left[ \log \frac{\pi_\star^\beta}{\pi} \right]$$

$$\leq 2\epsilon c_{\max} + \beta \mathbb{E}_\pi \left[ \log \frac{\pi}{\pi_{\text{ref}}} \right] - \beta \mathbb{E}_{\pi_\star^\beta} \left[ \log \frac{\pi}{\pi_{\text{ref}}} \right],$$

where the first inequality follows from Lemma 4 and the last inequality non-negativity of KL divergence:

$$-\beta \mathbb{E}_{\pi_\star^\beta} \left[ \log \frac{\pi_\star^\beta}{\pi} \right] = -\beta \mathbb{E}_{x \sim \rho} \left[ D_{\text{KL}} \left( \pi_\star^\beta(Y \mid x) \| \pi(Y \mid x) \right) \right] \leq 0.$$

Finally, we have:

$$\texttt{SubOpt}(\pi) \leq 2\epsilon c_{\max} + \beta \mathbb{E}_\pi \left[ \log \frac{\pi}{\pi_{\text{ref}}} \right] - \beta \mathbb{E}_{\pi_\star^\beta} \left[ \log \frac{\pi}{\pi_{\text{ref}}} \right]$$

$$= 2\epsilon c_{\max} + \beta \mathbb{E}_{x \sim \rho} \left[ \sum_{y \in \mathcal{Y}} \left( \pi(y \mid x) - \pi_\star^\beta(y \mid x) \right) \log \frac{\pi(y \mid x)}{\pi_\star^\beta(y \mid x)} \right]$$

$$\leq 2\epsilon c_{\max} + \beta \mathbb{E}_{x \sim \rho} \left[ \sum_{y \in \mathcal{Y}} \left| \pi(y \mid x) - \pi_\star^\beta(y \mid x) \right| \left| \log \frac{\pi(y \mid x)}{\pi_\star^\beta(y \mid x)} \right| \right]$$

$$\leq 2\epsilon c_{\max} + 2V_{\max} \mathbb{E}_{x \sim \rho} \left[ \left\| \pi(Y \mid x) - \pi_\star^\beta(Y \mid x) \right\|_{\text{TV}} \right]$$

$$\leq 2\epsilon(c_{\max} + V_{\max}),$$

where the third step uses triangle inequality and fourth step uses Assumption 4. Alternatively we can bound $\beta\mathbb{E}_{x\sim\rho}\left[\sum_{y\in\mathcal{Y}}\left|\pi(y\mid x) - \pi_\star^\beta(y\mid x)\right|\left|\log\frac{\pi(y\mid x)}{\pi_\star^\beta(y\mid x)}\right|\right]$ as follows,

$$\beta\mathbb{E}_{x\sim\rho}\left[\sum_{y\in\mathcal{Y}}\left|\pi(y\mid x) - \pi_\star^\beta(y\mid x)\right|\left|\log\frac{\pi(y\mid x)}{\pi_\star^\beta(y\mid x)}\right|\right] \leqslant$$

$$\leqslant 2V_{\max}\mathbb{E}_{x\sim\rho}\left[\left\|\pi(Y\mid x) - \pi_\star^\beta(Y\mid x)\right\|_{\mathrm{TV}}\right]$$

$$\leqslant 2\epsilon V_{\max},$$

Finally, $\texttt{Reg}_T(\pi) = \sum_{t=1}^T \texttt{SubOpt}(\pi) = \texttt{SubOpt}(\pi)T \leqslant 2\epsilon(c_{\max} + V_{\max})T.$ $\qquad\square$

The advantage of these results is that we can focus on bounding total variation of the learned policy with respect to $\pi_\star^\beta$ and the result for sub-optimality and regret for either regularized or unregularized objective follows.

### A.3 Leaning from Supervised Feedback

We recall that we are given a dataset of user-edits $\mathcal{D} = \{(x_i, y_i, y_i')\}_{i=1}^n$ where for every $i \in [n]$, we have $x_i \sim \rho(\cdot)$, $y_i \sim \pi_{\mathrm{ref}}(\cdot\mid x_i)$, and $y_i' \sim q(\cdot\mid x_i, y_i)$. The supervised fine-tuning (SFT) approach learns the following policy:

$$\hat{\pi}_{\mathrm{SUP}} = \arg\max_{\pi\in\Pi}\frac{1}{n}\sum_{i=1}^n \log\pi(y_i'\mid x_i).$$

**Theorem 4** (SFT Result Full). *Under Assumption 1, Assumption 2 and Assumption 4, we have that when $n \geqslant \frac{4(c_{max}+V_{max})^2\log(\ln|\Pi|/\delta)}{\epsilon^2}$ then with probability at least $1-\delta$:*

$$\texttt{SubOpt}(\hat{\pi}_{SUP}) \leqslant \epsilon + 2(c_{max} + V_{max})\left(\Delta_{mle}(n, \delta) + \min\left(\eta_{\max}\cdot D(\pi_{\mathrm{ref}}, \pi_\star^\beta), \bar{\eta}_{\max}\cdot D^{1/2}(\pi_{\mathrm{ref}}, \pi_\star^\beta)\right)\right)$$

*where $\eta_{\max} = 1 - \min_x\gamma_{min}(x)$ and $\bar{\eta}_{\max} = \sqrt{\mathbb{E}_{x\sim\rho}\left[(1-\gamma_{min}(x))^2\right]}$ for $\gamma_{min}(x)$ defined in Assumption 1 and*

$$D(\pi_{\mathrm{ref}}, \pi_\star^\beta) = \mathbb{E}_{x\sim\rho}\left[\left\|\pi_{\mathrm{ref}}(Y\mid x) - \pi_\star^\beta(Y\mid x)\right\|_{TV}\right].$$

*Proof.* The Bayes classifier of this log-likelihood problem is given by $q\circ\pi_{\mathrm{ref}} : \mathcal{X}\to\Delta(\mathcal{Y})$ where $q\circ\pi_{\mathrm{ref}}(y'\mid x) = \sum_{y\in\mathcal{Y}} q(y'\mid y, x)\pi_{\mathrm{ref}}(y\mid x)$. From Assumption 2 we have realizability and so using standard finite-sample guarantees for maximum likelihood (e.g., [23]) we have

$$\mathbb{E}_{x\sim\rho}\left[\left\|\hat{\pi}_{\mathrm{SUP}}(Y\mid x) - q\circ\pi_{\mathrm{ref}}(Y\mid x)\right\|_{\mathrm{TV}}\right] \leqslant \Delta_{\mathrm{mle}}(n, \delta) := \sqrt{\frac{1}{n}\log\left(\frac{\ln|\Pi|}{\delta}\right)} \qquad (17)$$

with probability at least $1-\delta$ for any fixed $\delta\in(0,1)$. Using triangle inequality we have:

$$\mathbb{E}_{x\sim\rho}\left[\left\|\hat{\pi}_{\mathrm{SUP}}(Y\mid x) - \pi_\star^\beta(Y\mid x)\right\|_{\mathrm{TV}}\right] \leqslant \mathbb{E}_{x\sim\rho}\left[\left\|\hat{\pi}_{\mathrm{SUP}}(Y\mid x) - q\circ\pi_{\mathrm{ref}}(Y\mid x)\right\|_{\mathrm{TV}}\right] +$$

$$\mathbb{E}_{x\sim\rho}\left[\left\|q\circ\pi_{\mathrm{ref}}(Y\mid x) - \pi_\star^\beta(Y\mid x)\right\|_{\mathrm{TV}}\right],$$

$$\leqslant \Delta_{\mathrm{mle}}(n, \delta) + \mathbb{E}_{x\sim\rho}\left[(1-\gamma_{\min}(x))\left\|\pi_{\mathrm{ref}}(Y\mid x) - \pi_\star^\beta(Y\mid x)\right\|_{\mathrm{TV}}\right],$$

where the last line uses Lemma 1 and Equation 17.

We compute two distinct bounds for the last term and consider the tightest among them:

$$\mathbb{E}_{x\sim\rho}\left[(1-\gamma_{\min}(x))\left\|\pi_{\mathrm{ref}}(Y\mid x) - \pi_\star^\beta(Y\mid x)\right\|_{\mathrm{TV}}\right]$$

$$\overset{(ii)}{\leqslant}\sqrt{\mathbb{E}_{x\sim\rho}\left[(1-\gamma_{\min}(x))^2\right]\cdot\mathbb{E}_{x\sim\rho}\left[\left\|\pi_{\mathrm{ref}}(Y\mid x) - \pi_\star^\beta(Y\mid x)\right\|_{\mathrm{TV}}^2\right]}$$

$$\overset{(ii)}{\leqslant}\sqrt{\mathbb{E}_{x\sim\rho}\left[(1-\gamma_{\min}(x))^2\right]\mathbb{E}_{x\sim\rho}\left[\left\|\pi_{\mathrm{ref}}(Y\mid x) - \pi_\star^\beta(Y\mid x)\right\|_{\mathrm{TV}}\right]}$$

where inequality $(i)$ holds because of Cauchy-Schwartz and $(ii)$ because the TV distance lies in $[0, 1]$. Moreover, the following inequality also holds:

$$\mathbb{E}_{x\sim\rho}\left[(1-\gamma_{\min}(x))\left\|\pi_{\mathrm{ref}}(Y\mid x)-\pi_\star^\beta(Y\mid x)\right\|_{\mathrm{TV}}\right] \leqslant$$
$$(1-\min_x\gamma_{\min}(x))\cdot\mathbb{E}_{x\sim\rho}\left[\left\|\pi_{\mathrm{ref}}(Y\mid x)-\pi_\star^\beta(Y\mid x)\right\|_{\mathrm{TV}}\right]$$

Thus we conclude that,

$$\mathbb{E}_{x\sim\rho}\left[\left\|\hat{\pi}_{\mathrm{SUP}}(Y\mid x)-\pi_\star^\beta(Y\mid x)\right\|_{\mathrm{TV}}\right] \leqslant \Delta_{\mathrm{mle}}(n,\delta)+$$
$$\min\left((1-\min_x\gamma_{\min}(x))D(\pi_{\mathrm{ref}},\pi_\star^\beta),\sqrt{\mathbb{E}_{x\sim\rho}\left[(1-\gamma_{\min}(x))^2\right]}D^{1/2}(\pi_{\mathrm{ref}},\pi_\star^\beta)\right)$$

We can use Lemma 4 or Lemma 5 to get bounds on sub-optimality and regret. In particular, using Lemma 5 we get:

$$\mathtt{SubOpt}(\hat{\pi}_{\mathrm{SUP}}) \leqslant 2(c_{\max}+V_{\max})\left(\Delta_{\mathrm{mle}}(n,\delta)+\min\left(\eta_{\max}\cdot D(\pi_{\mathrm{ref}},\pi_\star^\beta),\bar{\eta}_{\max}\cdot D^{1/2}(\pi_{\mathrm{ref}},\pi_\star^\beta)\right)\right),$$

$$\mathtt{Reg}_T(\hat{\pi}_{\mathrm{SUP}}) \leqslant 2(c_{\max}+V_{\max})\left(\Delta_{\mathrm{mle}}(n,\delta)+\min\left(\eta_{\max}\cdot D(\pi_{\mathrm{ref}},\pi_\star^\beta),\bar{\eta}_{\max}\cdot D^{1/2}(\pi_{\mathrm{ref}},\pi_\star^\beta)\right)\right)T.$$

where recall that $\eta_{\max}=1-\min_x\gamma_{\min}(x)$ and $\bar{\eta}_{\max}=\sqrt{\mathbb{E}_{x\sim\rho}\left[(1-\gamma_{\min}(x))^2\right]}$.

Finally, setting $n=\frac{4(c_{\max}+V_{\max})^2\log(\ln|\Pi|/\delta)}{\epsilon^2}$ we get the desired result. $\qquad\square$

### A.4 Theoretical Analysis with Preference Feedback

Given the user edit data $\mathcal{D}=\{(x_i,y_i,y_i')\}_{i=1}^n$, we first create a preference learning dataset $\mathcal{D}_{\mathrm{pref}}=\{(x_i,\tilde{y}_i,\tilde{y}_i',z_i)\}_{i=1}^n$ as follows: for every $i\in[n]$, we sample $z_i\in\mathtt{Unf}(\{\pm1\})$ and if $z_i=1$ then we define $\tilde{y}_i=y_i$ and $\tilde{y}_i'=y_i'$, otherwise, we define $\tilde{y}_i=y_i'$ and $\tilde{y}_i'=y_i$.

Given the LLM policy class $\Pi$, we induce a binary classifier class $\mathcal{C}=\{f_\pi:(x,\tilde{y},\tilde{y}')\to\Delta(\{\pm1\})\mid\forall\pi\in\Pi\}$ as:

$$\mathcal{C}=\{f_\pi:(x,\tilde{y},\tilde{y}')\to\Delta(\{\pm1\})\mid\forall\pi\in\Pi\},\quad\text{where}$$
$$\forall\pi\in\Pi,\quad f_\pi(Z=z\mid x,\tilde{y},\tilde{y}')=\sigma\left(z\log\frac{\pi(\tilde{y}'\mid x)}{\pi_{\mathrm{ref}}(\tilde{y}'\mid x)}-z\log\frac{\pi(\tilde{y}\mid x)}{\pi_{\mathrm{ref}}(\tilde{y}\mid x)}\right). \tag{18}$$

The size of $\mathcal{C}$ is the same as the size of the policy class $\Pi$.

We define the empirical log-likelihood $\ell_{\mathrm{PREF}}(\pi)$ for a policy $\pi\in\Pi$ as:

$$\ell_{\mathrm{PREF}}(\pi)=\frac{1}{n}\sum_{i=1}^n\log f_\pi\left(z_i\mid x_i,\tilde{y}_i,\tilde{y}_i'\right). \tag{19}$$

Using the construction of $\mathcal{D}_{\mathrm{pref}}$ from $\mathcal{D}$ we can express $\ell_{\mathrm{PREF}}(\pi)$ as:

$$\ell_{\mathrm{PREF}}(\pi)=\frac{1}{n}\sum_{i=1}^n\log\sigma\left(\log\frac{\pi(y_i'\mid x_i)}{\pi_{\mathrm{ref}}(y_i'\mid x_i)}-\log\frac{\pi(y_i\mid x_i)}{\pi_{\mathrm{ref}}(y_i\mid x_i)}\right), \tag{20}$$

which is the standard DPO loss without $\beta$ factor. Our theoretical analysis uses this variant of DPO without $\beta$ since in our analysis $\beta$ is a given value with respect to which Assumption 1 holds. In our experiments, we use the standard version of DPO with the $\beta$ term. Given the model class $\mathcal{C}$, we maximize the log-likelihood:

$$\hat{\pi}_{\mathrm{PREF}}=\arg\max_{f_\pi\in\mathcal{C}}\ell_{\mathrm{PREF}}(\pi). \tag{21}$$

Each datapoint $(x_i,\tilde{y}_i,\tilde{y}_i',z_i)\in\mathcal{D}_{\mathrm{pref}}$ is sampled IID from a distribution and let this be denoted by $D_{\mathrm{pref}}(x,\tilde{y},\tilde{y}',z)$. From the construction of $\mathcal{D}_{\mathrm{pref}}$ we have:

$$D_{\mathrm{pref}}(x,\tilde{y},\tilde{y}',z)=\begin{cases}\frac{1}{2}\rho(x)\pi_{\mathrm{ref}}(\tilde{y}\mid x)q(\tilde{y}'\mid x,\tilde{y}), & z=1\\ \frac{1}{2}\rho(x)\pi_{\mathrm{ref}}(\tilde{y}'\mid x)q(\tilde{y}\mid x,\tilde{y}'), & z=-1\end{cases} \tag{22}$$

Here we show in detail our calculations from Section 4 of how the distribution $\mathcal{D}_{\mathrm{pref}}(z = 1 \mid x, \tilde{y}, \tilde{y}')$ satisfies the Bradley-Terry Assumption. As we are using a variant of DPO without $\beta$, we will instead use a variant of Bradley-Terry Distribution as well which has a $\beta$ temperature.

**Lemma 6** (Satisfying Bradley-Terry Distribution). *Under Assumption 1, the probability of preferring $\tilde{y}'$ over $\tilde{y}$, i.e., $\tilde{y} \preceq \tilde{y}'$, in our preference dataset $\mathcal{D}_{pref}$ satisfies the Bradley-Terry assumption with respect to the cost function c. Formally, for every $x \in \mathrm{supp}\, \rho, \tilde{y}, \tilde{y}' \in \mathcal{Y}$ we have:*

$$\mathcal{D}_{pref}(z = 1 \mid x, \tilde{y}, \tilde{y}') = \sigma\left(\frac{c(x, y) - c(x, \tilde{y}')}{\beta}\right),$$

*where $\sigma(t) = \frac{1}{1+\exp(-t)}$ is the sigmoid function.*

*Proof.* The probability of the event $\tilde{y} \preceq \tilde{y}'$ is given by $\mathcal{D}_{\mathrm{pref}}(z = 1 \mid x, \tilde{y}, \tilde{y}')$ which is also the Bayes' classifier of our classification problem. Starting from Bayes's theorem we get:

$$
\begin{aligned}
\mathcal{D}_{\mathrm{pref}}(z = 1 \mid x, \tilde{y}, \tilde{y}') &= \frac{\mathcal{D}_{\mathrm{pref}}(z = 1, x, \tilde{y}, \tilde{y}')}{\mathcal{D}_{\mathrm{pref}}(z = 1, x, \tilde{y}, \tilde{y}') + \mathcal{D}_{\mathrm{pref}}(z = -1, x, \tilde{y}, \tilde{y}')}, \\
&= \frac{\pi_{\mathrm{ref}}(\tilde{y} \mid x) q(\tilde{y}' \mid x, \tilde{y})}{\pi_{\mathrm{ref}}(\tilde{y} \mid x) q(\tilde{y}' \mid x, \tilde{y}) + \pi_{\mathrm{ref}}(\tilde{y}' \mid x) q(\tilde{y} \mid x, \tilde{y}')}, \quad \text{(using Equation 22),} \\
&= \frac{\pi_{\mathrm{ref}}(\tilde{y} \mid x) \pi_\star^\beta(\tilde{y}' \mid x)}{\pi_{\mathrm{ref}}(\tilde{y} \mid x) \pi_\star^\beta(\tilde{y}' \mid x) + \pi_{\mathrm{ref}}(\tilde{y}' \mid x) \pi_\star^\beta(\tilde{y} \mid x)}, \quad \text{(using Assumption 1),} \\
&= \frac{\exp(-c(x, \tilde{y}')/\beta)}{\exp(-c(x, \tilde{y}')/\beta) + \exp(-c(x, \tilde{y})/\beta)}, \quad \text{(using Lemma 3)} \\
&= \sigma\left(\frac{c(x, \tilde{y}) - c(x, \tilde{y}')}{\beta}\right).
\end{aligned}
$$

$\square$

From Assumption 2 we have $\pi_\star^\beta \in \Pi$ which gives

$$
\begin{aligned}
f_{\pi_\star^\beta}(Z = 1 \mid x, \tilde{y}, \tilde{y}') &= \sigma\left(\beta \log \frac{\pi_\star^\beta(\tilde{y}' \mid x)}{\pi_{\mathrm{ref}}(\tilde{y}' \mid x)} - \beta \log \frac{\pi_\star^\beta(\tilde{y} \mid x)}{\pi_{\mathrm{ref}}(\tilde{y} \mid x)}\right) \\
&= \sigma\left(\frac{c(x, \tilde{y}) - c(x, \tilde{y}')}{\beta}\right).
\end{aligned}
$$

This means that we have realizability for the classification problem in Equation 19. We can then use standard MLE guarantees to state our next result.

**Lemma 7** (MLE Guarantee). *For any $\delta \in (0, 1)$, under Assumption 2 we have:*

$$\mathbb{E}_{(x, \tilde{y}, \tilde{y}') \sim D_{pref}}\left[\left\|f_{\hat{\pi}_{PREF}}(Z \mid x, \tilde{y}, \tilde{y}') - D_{pref}(Z \mid x, \tilde{y}, \tilde{y}')\right\|_{TV}\right] \leqslant \sqrt{\frac{2}{n} \ln \frac{|\Pi|}{\delta}}, \quad (23)$$

*where $D_{pref}(x, \tilde{y}, \tilde{y}')$ is the marginal distribution of Equation 22 and given by $D_{pref}(x, \tilde{y}, \tilde{y}') = \frac{\rho(x)}{2}\left(\pi_{\mathrm{ref}}(\tilde{y} \mid x) q(\tilde{y}' \mid x, \tilde{y}) + \pi_{\mathrm{ref}}(\tilde{y}' \mid x) q(\tilde{y} \mid x, \tilde{y}')\right)$.*

*Proof.* This is a standard MLE bound under realizability. Proof follows from using Theorem 21 of Agarwal et al. [2] and applying Chernoff's bound $\mathbb{E}[\sqrt{X^2}] \leqslant \sqrt{\mathbb{E}[X^2]}$. $\square$

Using $\|p - q\|_{\mathrm{TV}} = \frac{1}{2}\|p - q\|_1$ for two finite value distributions $p, q$ we get:

$$
\begin{aligned}
&\left\|f_{\hat{\pi}_{\mathrm{PREF}}}(Z \mid x, \tilde{y}, \tilde{y}') - D_{\mathrm{pref}}(Z \mid x, \tilde{y}, \tilde{y}')\right\|_{\mathrm{TV}} \\
&= \left|f_{\hat{\pi}_{\mathrm{PREF}}}(Z = 1 \mid x, \tilde{y}, \tilde{y}') - D_{\mathrm{pref}}(Z = 1 \mid x, \tilde{y}, \tilde{y}')\right|.
\end{aligned} \quad (24)
$$

For convenience, we define a notion of implicit cost $\hat{c}(x, y)$ given by:

$$\hat{\pi}_{\mathrm{PREF}}(y \mid x) = W(x) \pi_{\mathrm{ref}}(y \mid x) \exp(-\hat{c}(x, y)/\beta), \quad (25)$$

where $W(x) = \sum_{y' \in \mathcal{Y}} \pi_{\text{ref}}(y' \mid x) \exp(-\hat{c}(x, y')/\beta)$ is the normalization constant. As $\hat{c} \to c$, we have $\hat{\pi}_{\text{PREF}} \to \pi_\star^\beta$. Rewriting Equation 25, we get:

$$\hat{c}(x, y) = -\beta \log \frac{\hat{\pi}_{\text{PREF}}(y \mid x)}{\pi_{\text{ref}}(y \mid x)} + \beta \log W(x) \tag{26}$$

which gives

$$f_{\hat{\pi}_{\text{PREF}}}(Z = 1 \mid x, \tilde{y}, \tilde{y}') = \sigma \left( \frac{\hat{c}(x, \tilde{y}) - \hat{c}(x, \tilde{y}')}{\beta} \right). \tag{27}$$

Using Equations 24 and 27 and Lemma 6 we can simplify Equation 23 to:

$$\mathbb{E}_{(x, \tilde{y}, \tilde{y}') \sim D_{\text{pref}}} \left[ \left| \sigma \left( \frac{\hat{c}(x, \tilde{y}) - \hat{c}(x, \tilde{y}')}{\beta} \right) - \sigma \left( \frac{c(x, \tilde{y}) - c(x, \tilde{y}'))}{\beta} \right) \right| \right] \leqslant \sqrt{\frac{2}{n} \ln \frac{|\Pi|}{\delta}}. \tag{28}$$

To further simplify this result, we will need certain properties of the sigmoid function (Lemma 13 and Lemma 14). We will also assume Assumption 4.

Let $Z_1 = {(\hat{c}(x, \tilde{y}) - \hat{c}(x, \tilde{y}'))}/{\beta}$ and $Z_2 = {(c(x, \tilde{y}) - c(x, \tilde{y}'))}/{\beta}$. We have $|Z_1| \leqslant {2V_{\max}}/{\beta}$ from Assumption 4 and $|Z_2| \leqslant c_{\max}$ since $c(x, y) \in [0, c_{\max}]$. Then starting from Equation 28 we get:

$$\left| \sigma \left( \hat{c}(x, \tilde{y}) - \hat{c}(x, \tilde{y}') \right) - \sigma(c(x, \tilde{y}) - c(x, \tilde{y}')) \right|$$
$$= |\sigma(Z_1) - \sigma(Z_2)|$$
$$\geqslant \min\{\sigma'(Z_1), \sigma'(Z_2)\} |Z_1 - Z_2|, \qquad \text{(using Lemma 13)},$$
$$\geqslant \min \left\{ \sigma'(2V_{\max}/\beta), \sigma'(c_{\max}) \right\} |Z_1 - Z_2| \qquad \text{(using Lemma 14)}$$

This gives us:

$$\mathbb{E}_{(x, \tilde{y}, \tilde{y}') \sim D_{\text{pref}}} \left[ \left| \left( \hat{c}(x, \tilde{y}) - c(x, \tilde{y}') \right) - \left( c(x, \tilde{y}) - c(x, \tilde{y}') \right) \right| \right] \leqslant \epsilon_n, \tag{29}$$

where $\epsilon_n = \frac{\beta}{\min\{\sigma'(2V_{\max}/\beta), \sigma'(c_{\max})\}} \sqrt{\frac{2}{n} \ln \frac{|\Pi|}{\delta}}$.

Using Lemma 3 we have:

$$c(x, \tilde{y}) = -\beta \log \frac{\pi_\star^\beta(\tilde{y} \mid x)}{\pi_{\text{ref}}(\tilde{y} \mid x)} + \beta \log W^\star(x). \tag{30}$$

Plugging in values of $c$ from Equation 30 and $\hat{c}$ from Equation 26 in Equation 29 we get:

$$\mathbb{E}_{D_{\text{pref}}} \left[ \left| \left( \beta \log \frac{\hat{\pi}_{\text{PREF}}(\tilde{y} \mid x)}{\pi_{\text{ref}}(\tilde{y} \mid x)} - \beta \log \frac{\hat{\pi}_{\text{PREF}}(\tilde{y}' \mid x)}{\pi_{\text{ref}}(\tilde{y}' \mid x)} \right) - (\beta \log \frac{\pi_\star^\beta(\tilde{y} \mid x)}{\pi_{\text{ref}}(\tilde{y} \mid x)} - \beta \log \frac{\pi_\star^\beta(\tilde{y}' \mid x)}{\pi_{\text{ref}}(\tilde{y}' \mid x)} \right| \right] \leqslant \epsilon_n,$$

which simplifies to

$$\mathbb{E}_{(x, \tilde{y}, \tilde{y}') \sim D_{\text{pref}}} \left[ \left| \beta \log \frac{\hat{\pi}_{\text{PREF}}(\tilde{y} \mid x)}{\pi_\star^\beta(\tilde{y} \mid x)} - \beta \log \frac{\hat{\pi}_{\text{PREF}}(\tilde{y}' \mid x)}{\pi_\star^\beta(\tilde{y}' \mid x)} \right| \right] \leqslant \epsilon_n,$$

We now invoke the concentrability assumption (Assumption 3) which we restate below for convenience:

**Assumption 3.** *[Preference Concentrability] For any $\pi \in \Pi$ we assume:*

$$\frac{\mathbb{E}_{(x, \tilde{y}, \tilde{y}') \sim Q_\pi} \left[ \left| \beta \log \frac{\pi(\tilde{y}'|x)}{\pi_\star^\beta(\tilde{y}'|x)} - \beta \log \frac{\pi(\tilde{y}|x)}{\pi_\star^\beta(\tilde{y}|x)} \right| \right]}{\mathbb{E}_{(x, \tilde{y}, \tilde{y}') \sim Q_{\pi_{\text{ref}}}} \left[ \left| \beta \log \frac{\pi(\tilde{y}'|x)}{\pi_\star^\beta(\tilde{y}'|x)} - \beta \log \frac{\pi(\tilde{y}|x)}{\pi_\star^\beta(\tilde{y}|x)} \right| \right]} \leqslant C_{\text{PREF}},$$

*where $Q_\pi(x, \tilde{y}, \tilde{y}') = \frac{1}{2} \rho(x) \left( \pi(\tilde{y}' \mid x)\pi^\star(\tilde{y} \mid x) + \pi^\star(\tilde{y}' \mid x)\pi(\tilde{y} \mid x) \right)$.*

to get:

$$\mathbb{E}_{(x,\tilde{y},\tilde{y}')\sim Q_{\hat{\pi}_{\text{PREF}}}}\left[\left|\beta\log\frac{\hat{\pi}_{\text{PREF}}(\tilde{y}'\mid x)}{\pi_\star^\beta(\tilde{y}'\mid x)}-\beta\log\frac{\hat{\pi}_{\text{PREF}}(\tilde{y}\mid x)}{\pi_\star^\beta(\tilde{y}\mid x)}\right|\right]\leqslant C_{\text{PREF}}\cdot\epsilon_n, \tag{31}$$

where we used the fact that $D_{\text{pref}}=Q_{\pi_{\text{ref}}}$. Concentrability style assumptions are common in analysis of RL algorithms [14, 44]. Our assumption is similar to the one in Theorem 1 of Rosset et al., [44]. the main difference is that our assumption is designed for learning from user-edits while that of Rosset et al., is designed for iterative learning with preference feedback.

Simplifying Equation 31 gives:

$$\mathbb{E}_{(x,\tilde{y},\tilde{y}')\sim Q_{\hat{\pi}_{\text{PREF}}}}\left[\left|\log\frac{\hat{\pi}_{\text{PREF}}(\tilde{y}'\mid x)\pi_\star^\beta(\tilde{y}\mid x)}{\pi_\star^\beta(\tilde{y}'\mid x)\hat{\pi}_{\text{PREF}}(\tilde{y}\mid x)}\right|\right]\leqslant\frac{C_{\text{PREF}}}{\beta}\cdot\epsilon_n \tag{32}$$

We lower bound the left hand side of this assumption:

$$\mathbb{E}_{(x,\tilde{y},\tilde{y}')\sim Q_{\hat{\pi}_{\text{PREF}}}}\left[\left|\log\frac{\hat{\pi}_{\text{PREF}}(\tilde{y}'\mid x)_\star^\beta(\tilde{y}\mid x)}{\pi_\star^\beta(\tilde{y}'\mid x)\hat{\pi}_{\text{PREF}}(\tilde{y}\mid x)}\right|\right],$$

$$\geqslant\mathbb{E}_{(x,\tilde{y},\tilde{y}')\sim Q_{\hat{\pi}_{\text{PREF}}}}\left[\frac{\left|\frac{\hat{\pi}_{\text{PREF}}(\tilde{y}'\mid x)\pi_\star^\beta(\tilde{y}\mid x)}{\pi_\star^\beta(\tilde{y}'\mid x)\hat{\pi}_{\text{PREF}}(\tilde{y}\mid x)}-1\right|}{\frac{\hat{\pi}_{\text{PREF}}(\tilde{y}'\mid x)\pi_\star^\beta(\tilde{y}\mid x)}{\pi_\star^\beta(\tilde{y}'\mid x)\hat{\pi}_{\text{PREF}}(\tilde{y}\mid x)}+1}\right],\quad\text{(using }|\log(t)|\geqslant\frac{|t-1|}{t+1}\text{ for }t>0\text{ from Lemma 15)}$$

$$=\mathbb{E}_{(x,\tilde{y},\tilde{y}')\sim Q_{\hat{\pi}_{\text{PREF}}}}\left[\frac{\left|\hat{\pi}_{\text{PREF}}(\tilde{y}'\mid x)\pi_\star^\beta(\tilde{y}\mid x)-\pi_\star^\beta(\tilde{y}'\mid x)\hat{\pi}_{\text{PREF}}(\tilde{y}\mid x)\right|}{\hat{\pi}_{\text{PREF}}(\tilde{y}'\mid x)\pi_\star^\beta(\tilde{y}\mid x)+\pi_\star^\beta(\tilde{y}'\mid x)\hat{\pi}_{\text{PREF}}(\tilde{y}\mid x)}\right],$$

$$=\frac{1}{2}\mathbb{E}_{x\sim\rho}\left[\sum_{\tilde{y},\tilde{y}'\in\mathcal{Y}}\left|\hat{\pi}_{\text{PREF}}(\tilde{y}'\mid x)\pi_\star^\beta(\tilde{y}\mid x)-\pi_\star^\beta(\tilde{y}'\mid x)\hat{\pi}_{\text{PREF}}(\tilde{y}\mid x)\right|\right],\quad\text{(using the definition of }Q_{\hat{\pi}_{\text{PREF}}})$$

$$\geqslant\frac{1}{2}\mathbb{E}_{x\sim\rho}\left[\sum_{\tilde{y}\in\mathcal{Y}}\left|\sum_{\tilde{y}'\in\mathcal{Y}}\hat{\pi}_{\text{PREF}}(\tilde{y}'\mid x)\pi_\star^\beta(\tilde{y}\mid x)-\pi_\star^\beta(\tilde{y}'\mid x)\hat{\pi}_{\text{PREF}}(\tilde{y}\mid x)\right|\right],\quad\text{(triangle inequality)}$$

$$=\mathbb{E}_{x\sim\rho}\left[\left\|\pi_{\text{ref}}(Y\mid x)-\pi_\star^\beta(Y\mid x)\right\|_{\text{TV}}\right]$$

Combining this all we get:

$$\mathbb{E}_{x\sim\rho}\left[\left\|\pi_{\text{ref}}(Y\mid x)-\pi_\star^\beta(Y\mid x)\right\|_{\text{TV}}\right]\leqslant\frac{C_{\text{PREF}}}{\min\left\{\sigma'(2V_{\max}/\beta),\sigma'(c_{\max})\right\}}\sqrt{\frac{2}{n}\ln\frac{|\Pi|}{\delta}} \tag{33}$$

We can wrap up the proof with the application of Lemma 5.

**Theorem 5** (DPO Result). *Under Assumption 1, Assumption 2, and Assumption 3, Assumption 4 we have that when $n\geqslant\frac{8C_{\text{PREF}}^2(c_{max}+V_{max})^2\ln\frac{|\Pi|}{\delta}}{\min\{\sigma'(2V_{\max}/\beta),\sigma'(-c_{max})\}^2\epsilon^2}$ then with probability at least $1-\delta$:*

$$\texttt{SubOpt}(\hat{\pi}_{PREF})\leqslant\epsilon. \tag{34}$$

*Proof.* We set $n$ such that $\frac{\epsilon}{2(c_{max}+V_{max})}=\frac{C_{\text{PREF}}}{\min\{\sigma'(2V_{\max}/\beta),\sigma'(c_{\max})\}}\sqrt{\frac{2}{n}\ln\frac{|\Pi|}{\delta}}$ which together with Equation 33 and Lemma 5 gives us $\texttt{SubOpt}(\hat{\pi}_{\text{PREF}})\leqslant\epsilon$ and $\texttt{Reg}_T(\hat{\pi}_{\text{PREF}})\leqslant\epsilon T$. $\square$

We can simplify Theorem 4 and Theorem 5 if we define $V_{\max}$ to be the max of the quantity defined in Assumption 4 and $c_{\max}$.

## A.5 Learning from Cost

Throughout this section, we will assume that the cost functions are bounded.

**Assumption 5.** *The cost functions $f \in \mathcal{F}$ satisfy $|f(x, y)| \leqslant c_{max}$ for some $c_{max} > 0$ for all $x \in \mathcal{X}$ and $y \in \mathcal{Y}$.*

Provided the true cost function satisfies $c \in \mathcal{F}$ where $\mathcal{F}$ is a class of functions from the space of pairs $\mathcal{X} \times \mathcal{Y}$ to the real numbers we can estimate the mean cost function by least squares regression on the observed costs gathered by $\pi_{\mathrm{ref}}$.

$$\hat{f} = \arg\min_{f \in \mathcal{F}} \frac{1}{n} \sum_{i=1}^{n} (f(x_i, y_i) - c_i)^2 \,,$$

The estimator $\hat{f}$ satisfies,

**Lemma 8.** *Let and $\delta \in (0, 1)$ and $\{(x_i, y_i, c_i)\}_{i=1}^{n}$ be a dataset of prompt answers and costs. If Assumption 5 holds then*

$$\mathbb{E}_{x \sim \rho, y \sim \pi_{\mathrm{ref}}(\cdot|x)} \left[ (\hat{f}(x, y) - c(x, y))^2 \right] \leqslant b \cdot c_{max}^2 \log \left( \frac{|\mathcal{F}|}{\delta} \right)$$

*and defining the confidence set of plausible cost functions as*

$$\hat{\mathcal{F}} = \left\{ f \text{ s.t. } \sum_{i=1}^{n} \left( f(x_i, y_i) - \hat{f}(x_i, y_i) \right)^2 \leqslant b \cdot c_{max}^2 \log \left( \frac{|\mathcal{F}|}{\delta} \right) \right\} \tag{35}$$

*for a universal constant $b \geqslant 1$. We have $c \in \hat{\mathcal{F}}$ and $\mathbb{E}_{x \sim \rho, y \sim \pi_{\mathrm{ref}}(\cdot|x)} \left[ (f(x, y) - c(x, y))^2 \right] \leqslant b \cdot c_{max}^2 \log \left( \frac{|\mathcal{F}|}{\delta} \right)$ for all $f \in \hat{\mathcal{F}}$ with probability at least $1 - \delta/2$.*

Throughout this section we will define $\mathcal{E}$ as the at least $1 - \delta/2$ probability event defined in Lemma 8.

Given cost function $f$ we introduce notation to define the policy optimizing the regularized objective where we add $f$ to the notation.

$$\pi_{\star}^{f,\beta} = \arg\min_{\pi \in \Pi} \mathbb{E}_{x \sim \rho, y \sim \pi(\cdot|x)} \left[ f(x, y) + \beta \log \frac{\pi(y \mid x)}{\pi_{\mathrm{ref}}(y \mid x)} \right]$$

and satisfies $\pi_{\star}^{f,\beta}(y|x) = \frac{\exp\left(-\frac{f(x,y)}{\beta}\right) \pi_{\mathrm{ref}}(y|x)}{Z(x)}$ where $Z(x) = \sum_{y'} \exp\left(-\frac{f(x,y')}{\beta}\right) \cdot \pi_{\mathrm{ref}}(y'|x)$.

**Pesimism**    As we have mentioned in the main, we consider the following pessimistic estimator of the cost function,

$$\bar{f}(x, y) = \max_{f \in \hat{\mathcal{F}}} f(x, y)$$

where $\hat{\mathcal{F}}$ is defined as in equation 35. It follows that $\hat{\pi}_{\mathrm{RL}}$ as defined in Equation 6 satisfies $\hat{\pi}_{\mathrm{RL}} = \pi_{\star}^{\bar{f},\beta}$.

Moreover, Assumption 5 implies,

$$\max_{x \in \mathcal{X}, y \in \mathcal{Y}} |\bar{f}(x, y)| \leqslant c_{\max}.$$

### A.5.1 Regularized Cost Analysis

**Proposition 6.** *For any $x \in \mathrm{support}(\rho)$,*

$$-c_{max} \overset{(a)}{\leqslant} \mathbb{E}_{y \sim \pi_{\star}^{f,\beta}(\cdot|x)} \left[ f(x, y) + \beta \cdot \log \left( \frac{\pi_{\star}^{f,\beta}(y \mid x)}{\pi_{\mathrm{ref}}(y \mid x)} \right) \right] \overset{(b)}{=} -\beta \cdot \log(Z(x)) \overset{(c)}{\leqslant} c_{max}. \tag{36}$$

*Proof.* Equality $(b)$ follows by substitution. We now proceed to prove the inequalities $(a)$ and $(c)$. Now observe that $Z(x) = \mathbb{E}_{y \sim \pi_{\text{ref}}(\cdot|x)} \left[ \exp\left(-f(x,y)/\beta\right) \right]$. Notice that $g(\gamma) = \beta \cdot \log\left(1/\gamma\right)$ is a convex function. Thus, Jensen's inequality implies,

$$g\left(\mathbb{E}_{y \sim \pi_{\text{ref}}(\cdot|x)} \left[ \exp\left(-f(x,y)/\beta\right) \right]\right) \overset{(i)}{\leqslant} \mathbb{E}_{y \sim \pi_{\text{ref}}(\cdot X)} \left[ \beta \cdot \log\left(\exp\left(f(x,y)/\beta\right)\right) \right]$$
$$= \mathbb{E}_{y \sim \pi_{\text{ref}}(\cdot|x)} \left[ f(x,y) \right]$$
$$\overset{(ii)}{\leqslant} c_{\max}.$$

where inequality $(i)$ follows by Jensen and $(ii)$ by Assumption 5. To derive a lower bound we need to upper bound $\mathbb{E}_{y \sim \pi_{\text{ref}}(\cdot|x)} \left[ \exp\left(-f(x,y)/\beta\right) \right]$. Assumption 5 implies,

$$Z(x) = \mathbb{E}_{y \sim \pi_{\text{ref}}(\cdot|x)} \left[ \exp\left(-f(x,y)/\beta\right) \right] \leqslant \exp\left(c_{\max}/\beta\right)$$

Thus,

$$\beta \cdot \log\left(1/Z(x)\right) \geqslant -c_{\max}.$$

This concludes the proof. $\qquad\qquad\square$

Proposition 6 implies that

$$\pi_\star^{f,\beta} \in \Pi(f) = \left\{ \pi \text{ s.t. } \left| \mathbb{E}_{y \sim \pi(\cdot|x)} \left[ f(x,y) + \beta \cdot \log\left( \frac{\pi(y|x)}{\pi_{\text{ref}}(y|x)} \right) \right] \right| \leqslant c_{\max} \ \ \forall x \in \text{support}(\rho) \right\}.$$
$$(37)$$

For the next few results we condition on the event $\mathcal{E}$. Because the function $\bar{f}$ is not in the class of cost functions our results will depend on the Eluder dimension of $\mathcal{F}$. Pessimism allows us to prove bounds depending on the concentrability coefficient of the $\pi_{\text{ref}}$ sampling distribution. Assuming access to a new prompt dataset $\{\bar{x}_i\}_{i=1}^n$ we analyze the following algorithm,

$$\hat{\pi}_{\text{RL}} = \underset{\pi \in \Pi(\bar{f})}{\text{argmin}} \sum_{i=1}^n \mathbb{E}_{y \sim \pi(\cdot|\bar{x}_i)} \left[ \bar{f}(\bar{x}_i, y) + \beta \cdot \log\left( \pi(y|\bar{x}_i)/\pi_{\text{ref}}(y|\bar{x}_i) \right) \right].$$

where

$$\Pi(\bar{f}) = \left\{ \pi \text{ s.t. } \left| \mathbb{E}_{y \sim \pi(\cdot|x)} \left[ \bar{f}(x,y) + \beta \log\left( \frac{\pi(y|x)}{\pi_{\text{ref}}(y|x)} \right) \right] \right| \leqslant c_{\max} \ \ \forall x \in \text{support}(\rho) \right\}.$$

Equation 37 implies $\pi_\star^{\bar{f},\beta} \in \Pi(\bar{f})$.

In this section we'll write the sample complexity of our algorithms in terms of the Eluder dimension of $\mathcal{F}$. This is a measure of statistical complexity of a function class first introduced by [45] and is based on the notion of $\epsilon$-independence defined as,

**Definition 1.** *($\epsilon$−dependence) Let $\mathcal{G}$ be a scalar function class with domain $\mathcal{Z}$ and $\epsilon > 0$. An element $z \in \mathcal{Z}$ is $\epsilon$−dependent on $\{z_1, \cdots, z_n\} \subseteq \mathcal{Z}$ w.r.t. $\mathcal{G}$ if any pair of functions $g, g' \in \mathcal{G}$ satisfying $\sqrt{\sum_{i=1}^n (g(z_i) - g'(z_i))^2} \leqslant \epsilon$ also satisfies $g(z) - g'(z) \leqslant \epsilon$. Furthermore, $z \in \mathcal{Z}$ is $\epsilon$−independent of $\{z_1, \cdots, z_n\}$ w.r.t. $\mathcal{G}$ if it is not $\epsilon$−dependent on $\{z_1, \cdots, z_n\}$.*

The eluder dimension is defined as,

**Definition 2.** *($\epsilon$-eluder) Let $\mathcal{G}$ be a function class with support in a set $\mathcal{Z}$ and values in $\mathbb{R}$. The $\epsilon$−non monotone eluder dimension $\bar{d}_{\text{eluder}}(\mathcal{G}, \epsilon)$ of $\mathcal{G}$ is the length of the longest sequence of elements in $\mathcal{Z}$ such that every element is $\epsilon$−independent of its predecessors. Moreover, we define the $\epsilon$−eluder dimension $d_{\text{eluder}}(\mathcal{G}, \epsilon)$ as $d_{\text{eluder}}(\mathcal{G}, \epsilon) = \max_{\epsilon' \geqslant \epsilon} \bar{d}_{\text{eluder}}(\mathcal{G}, \epsilon)$.*

The following eluder dimension lemmas will also prove useful in proving our sample complexity bounds. The first one is a restatement of Proposition 3 from [45].

**Lemma 9.** *Let $\{x'_\ell, y'_\ell\}_{\ell=1}^n$ be an arbitrary sequence in $\mathcal{X} \times \mathcal{Y}$. When $\mathcal{E}$ holds ,*

$$\sum_{\ell=1}^n \mathbf{1} \left( |\bar{f}(x'_\ell, y'_\ell) - c(x'_\ell, y'_\ell)| > \epsilon \right) \leqslant \left( 1 + \frac{4b \cdot c_{max}^2 \log\left( \frac{|\mathcal{F}|}{\delta} \right)}{\epsilon^2} \right) \cdot d_{\text{eluder}}(\mathcal{F}, \epsilon).$$

The result above implies,

**Lemma 10.** *Let $\{x'_\ell, y'_\ell\}_{\ell=1}^n$ be an arbitrary sequence in $\mathcal{X} \times \mathcal{Y}$. When $\mathcal{E}$ holds ,*

$$\sum_{\ell=1}^n (\bar{f}(x'_\ell, y'_\ell) - c(x'_\ell, y'_\ell))^2 \leqslant 17b \cdot c_{max}^2 \log \left( \frac{|\mathcal{F}|}{\delta} \right) \cdot \log(n) \cdot d_{\text{eluder}}(\mathcal{F}, \tau).$$

*for the same unversal constant $b \geqslant 1$ as in Lemma 8.*

*Proof.* We use the same proof technique as in Lemma 2 from [45].Let $w_i = (\bar{f}(x'_\ell, y'_\ell) - c(x'_\ell, y'_\ell))^2$.

Let $i_1, \cdots, i_n$ be the permutation of indices 1 to $n$ such that $w_{i_1} \geqslant \cdots \geqslant w_{i_n}$.

Let $\tau > 0$ be a parameter to be specified later on. For $t \in [n]$ such that $w_{i_t} \geqslant \tau$ notice that,

$$t \leqslant \sum_{\ell=1}^n \mathbf{1}\left(|\bar{f}(x'_\ell, y'_\ell) - c(x'_\ell, y'_\ell)| > w_{i_t}\right) \overset{(i)}{\leqslant} \left( 1 + \frac{4b \cdot c_{\max}^2 \log \left( \frac{|\mathcal{F}|}{\delta} \right)}{w_{i_t}^2} \right) \cdot d_{\text{eluder}}(\mathcal{F}, w_{i_t})$$

$$\overset{(ii)}{\leqslant} \left( 1 + \frac{4b \cdot c_{\max}^2 \log \left( \frac{|\mathcal{F}|}{\delta} \right)}{w_{i_t}^2} \right) \cdot d_{\text{eluder}}(\mathcal{F}, \tau)$$

$$\overset{(iii)}{\leqslant} \frac{8b \cdot c_{\max}^2 \log \left( \frac{|\mathcal{F}|}{\delta} \right)}{w_{i_t}^2} \cdot d_{\text{eluder}}(\mathcal{F}, \tau)$$

where $(i)$ follows by Lemma 9 and $(ii)$ follows because $d_{\text{eluder}}(\mathcal{F}, \cdot)$ is monotonically decreasing in the second argument. Inequality $(iii)$ holds because $w_{i_1} \leqslant 4c_{\max}^2$.

And therefore, for any $t$ such that $w_{i_t} \geqslant \tau$ we have,

$$w_{i_t}^2 \leqslant \frac{8b \cdot c_{\max}^2 \log \left( \frac{|\mathcal{F}|}{\delta} \right)}{t} \cdot d_{\text{eluder}}(\mathcal{F}, \tau).$$

and therefore,

$$\sum_{\ell=1}^n w_{i_\ell}^2 \cdot \mathbf{1}(w_{i_\ell} \geqslant \tau) \leqslant \sum_{\ell=1}^n \frac{8b \cdot c_{\max}^2 \log \left( \frac{|\mathcal{F}|}{\delta} \right)}{\ell} \cdot d_{\text{eluder}}(\mathcal{F}, \tau) \leqslant 16b \cdot c_{\max}^2 \log \left( \frac{|\mathcal{F}|}{\delta} \right) \cdot \log(n) \cdot d_{\text{eluder}}(\mathcal{F}, \tau).$$

And finally,

$$\sum_{\ell=1}^n w_{i_\ell}^2 \cdot \mathbf{1}(w_{i_\ell} < \tau) \leqslant \tau^2 \cdot n.$$

And therefore we have,

$$\sum_{\ell=1}^n w_{i_\ell}^2 \leqslant 16b \cdot c_{\max}^2 \log \left( \frac{|\mathcal{F}|}{\delta} \right) \cdot \log(n) \cdot d_{\text{eluder}}(\mathcal{F}, \tau) + \tau^2 \cdot n.$$

Finally, setting $\tau = c_{\max}/n$ we get the desired result,

$$\sum_{\ell=1}^n w_{i_\ell}^2 \leqslant 17b \cdot c_{\max}^2 \log \left( \frac{|\mathcal{F}|}{\delta} \right) \cdot \log(n) \cdot d_{\text{eluder}}(\mathcal{F}, c_{\max}/n).$$

$\square$

Additionally our results relies on the following variant of Bernstein inequality for martingales, or Freedman's inequality [20], as stated in e.g., [1, 8].

**Lemma 11** (Simplified Freedman's inequality)**.** *Let $Z_1, ..., Z_T$ be a bounded martingale difference sequence with $|Z_\ell| \leqslant R$. For any $\delta' \in (0, 1)$, and $\eta \in (0, 1/R)$, with probability at least $1 - \delta'$,*

$$\sum_{\ell=1}^T Z_\ell \leqslant \eta \sum_{\ell=1}^T \mathbb{E}_{\ell-1}[Z_\ell^2] + \frac{\log(1/\delta')}{\eta}. \tag{38}$$

*where $\mathbb{E}_{\ell-1}[\cdot]$ is the conditional expectation[3] induced by conditioning on $Z_1, \cdots, Z_{\ell-1}$.*

---

[3]We will use this notation to denote conditional expectations throughout this work.

The following Lemma will prove useful in our results relating the in-distribution ($\pi_{\text{ref}}$) error of the pessimistic cost estimators w.r.t. the true cost function $c$.

**Lemma 12.** *The following inequality holds,*

$$\mathbb{P}\left(\left\{\mathbb{E}_{\pi_{\text{ref}}}\left[(\bar{f}(x,y)-c(x,y))^2\right] \leqslant \mathcal{O}\left(\frac{d_{\text{eluder}}(\mathcal{F},1/n)\log(|\mathcal{F}|/\delta)}{n}\right)\right\} \cap \mathcal{E}\right) \geqslant 1 - 3\delta/4.$$

*Proof.* Let $\{(x'_\ell, y'_\ell)\}_{\ell=1}^n$ a fresh set of samples from $\rho \times \pi_{\text{ref}}$ and consider the martingale difference sequence $Z_\ell^f = (f(x'_\ell, y'_\ell) - c(x'_\ell, y'_\ell))^2 - \mathbb{E}_{x\sim\rho, y\sim\pi_{\text{ref}}(\cdot|x)}\left[(f(x,y)-c(x,y))^2\right]$ indexed by $f \in \hat{\mathcal{F}}$.

It follows that $|Z_\ell^f| \leqslant 4c_{\max}^2$ for all $\ell \in [n]$ and all $f \in \hat{\mathcal{F}}$. Moreover,

$$\mathbb{E}_{\ell-1}\left[\left(Z_\ell^f\right)^2\right] \leqslant 4c_{\max}^2 \mathbb{E}_{\ell-1}\left[\left|Z_\ell^f\right|\right] \leqslant 8c_{\max}^2 \mathbb{E}_{x\sim\rho, y\sim\pi_{\text{ref}}(\cdot|x)}\left[(f(x,y)-c(x,y))^2\right].$$

Where the last inequality holds because,

$$\mathbb{E}_{\ell-1}\left[\left|Z_\ell^f\right|\right] \leqslant \mathbb{E}_{\ell-1}[(f(x'_\ell, y'_\ell) - c(x'_\ell, y'_\ell))^2] + \mathbb{E}_{x\sim\rho, y\sim\pi_{\text{ref}}(\cdot|x)}\left[(f(x,y)-c(x,y))^2\right]$$

$$= 2\mathbb{E}_{x\sim\rho, y\sim\pi_{\text{ref}}(\cdot|x)}\left[(f(x,y)-c(x,y))^2\right]. \tag{39}$$

Using Freedman's inequality from Lemma 11 we conclude when $\mathcal{E}$ holds,

$$\sum_{\ell=1}^n (f(x'_\ell, y'_\ell) - c(x'_\ell, y'_\ell))^2 \leqslant n \cdot \mathbb{E}_{x\sim\rho, y\sim\pi_{\text{ref}}(\cdot|x)}[(f(x,y)-c(x,y))^2]+$$

$$8\eta n c_{\max}^2 \mathbb{E}_{x\sim\rho, y\sim\pi_{\text{ref}}(\cdot|x)}\left[(f(x,y)-c(x,y))^2\right] + \frac{\log(2|\mathcal{F}|/\delta)}{\eta}$$

$$\overset{(i)}{\leqslant} (1 + 8\eta c_{\max}^2)bc_{\max}^2 \log\left(\frac{|\mathcal{F}|}{\delta}\right) + \frac{\log(2|\mathcal{F}|/\delta)}{\eta}$$

$$\overset{(ii)}{=} 2bc_{\max}^2 \log\left(\frac{|\mathcal{F}|}{\delta}\right) + 8c_{\max}^2 \log(2|\mathcal{F}|/\delta)$$

$$= \mathcal{O}\left(c_{\max}^2 \log(|\mathcal{F}|/\delta)\right) := b'c_{\max}^2 \log(|\mathcal{F}|/\delta). \tag{40}$$

simultaneously for all $f \in \hat{\mathcal{F}}$ with probability at least $1 - \delta/8$.

Where inequality $(i)$ holds because of conditioning on $\mathcal{E}$ and Lemma 8 and $(ii)$ follows by setting $\eta = \frac{1}{8c_{\max}^2}$.

Let $\mathcal{E}'$ be the event where $\mathcal{E}$ and 40 hold. In this case,

$$\sum_{\ell=1}^t (f(x'_\ell, y'_\ell) - c(x'_\ell, y'_\ell))^2 \leqslant b'c_{\max}^2 \log(|\mathcal{F}|/\delta)$$

for all $t \in [0, \cdots, n]$.

We consider a new martingale difference sequence $\{\tilde{Z}_\ell\}_{\ell=1}^n$ defined as

$$\tilde{Z}_\ell = \mathbb{E}_{\pi_{\text{ref}}}[(\bar{f}(x,y)-c(x,y))^2] - (\bar{f}(x'_\ell, y'_\ell) - c(x'_\ell, y'_\ell))^2.$$

The following bounds hold, $|\tilde{Z}_\ell| \leqslant 4c_{\max}^2$ for all $\ell \in [n]$ and all $f \in \hat{\mathcal{F}}$. Moreover,

$$\mathbb{E}_{\ell-1}\left[\left(\tilde{Z}_\ell\right)^2\right] \leqslant 4c_{\max}^2 \mathbb{E}_{\ell-1}\left[\left|\tilde{Z}_\ell\right|\right] \leqslant 8c_{\max}^2 \mathbb{E}_{x\sim\rho, y\sim\pi_{\text{ref}}(\cdot|x)}\left[(\bar{f}(x,y)-c(x,y))^2\right].$$

where the last inequality has the same derivation as inequality 39. Freedman's inequality (Lemma 11) implies,

$$n\mathbb{E}_{\pi_{\text{ref}}}[(\bar{f}(x,y)-c(x,y))^2] \leqslant \sum_{\ell=1}^n (\bar{f}(x'_\ell, y'_\ell) - c(x'_\ell, y'_\ell))^2+$$

$$8\eta n c_{\max}^2 \mathbb{E}_{\pi_{\text{ref}}}[(\bar{f}(x,y)-c(x,y))^2] + \frac{\log(8/\delta)}{\eta}$$

with probability at least $1 - \delta/8$. By setting $\eta = \frac{1}{16c_{\max}^2}$ we conclude,

$$\mathbb{E}_{\pi_{\mathrm{ref}}}[(\bar{f}(x,y) - c(x,y))^2] \leqslant \frac{2}{n}\sum_{\ell=1}^{n}(\bar{f}(x_\ell', y_\ell') - c(x_\ell', y_\ell'))^2 + \frac{32c_{\max}^2 \log(8/\delta)}{n} \qquad (41)$$

with probability at least $1 - \delta/8$.

Lemma 10 implies that when $\mathcal{E}$ holds,

$$\begin{aligned}
\mathbb{E}_{\pi_{\mathrm{ref}}}[(\bar{f}(x,y) - c(x,y))^2] &\leqslant \frac{34b \cdot c_{\max}^2 \log\left(\frac{|\mathcal{F}|}{\delta}\right) \cdot \log(n) \cdot d_{\mathrm{eluder}}(\mathcal{F}, c_{\max}/n)}{n} + \frac{32c_{\max}^2 \log(8/\delta)}{n} \\
&\leqslant \frac{136b \cdot c_{\max}^2 \log\left(\frac{|\mathcal{F}|}{\delta}\right) \cdot \log(n) \cdot d_{\mathrm{eluder}}(\mathcal{F}, c_{\max}/n)}{n} \\
&\leqslant \mathcal{O}\left(\frac{c_{\max}^2 \log\left(\frac{|\mathcal{F}|}{\delta}\right) \cdot \log(n) \cdot d_{\mathrm{eluder}}(\mathcal{F}, c_{\max}/n)}{n}\right)
\end{aligned}$$

Combining $\mathcal{E}$ and the conitions for equations 40 and 41, the desired result holds with probability at least $1 - \delta/2 - \delta/8 - \delta/8$.

$\square$

**Theorem 7.** *The policy $\hat{\pi}_{RL}$ satisfies,*

$$J_\beta(\hat{\pi}_{RL}) \leqslant J_\beta(\pi_\star^{c,\beta}) + \mathcal{O}\left(\sqrt{\frac{\log(|\Pi|/\delta)}{n}} + \bar{C}_\star \cdot \sqrt{\frac{\log(n)d_{\mathrm{eluder}}(\mathcal{F}, c_{max}/n)\log(|\mathcal{F}|/\delta)}{n}}\right).$$

*with probability at least $1 - \delta$ where $\bar{C}_\star = \sqrt{\mathbb{E}_{\pi_{\mathrm{ref}}}\left[\left(\frac{\pi_{\mathrm{RL}}^{c,\beta}(y|x)}{\pi_{\mathrm{ref}}(y|x)}\right)^2\right]}$ is the optimal policy concentrability coefficient.*

*Proof.* Throughout this proof we will condition on the event $\mathcal{E}$. Since $\pi_\star^{\bar{f},\beta} \in \Pi(\bar{f})$ and by definition of $\hat{\pi}_{\mathrm{RL}}$,

$$\frac{1}{n}\sum_{i=1}^{n}\mathbb{E}_{y\sim\hat{\pi}_{\mathrm{RL}}(\cdot|\bar{x}_i)}\left[\bar{f}(\bar{x}_i, y) + \beta\log\left(\hat{\pi}_{\mathrm{RL}}(y|\bar{x}_i)/\pi_{\mathrm{ref}}(y|\bar{x}_i)\right)\right] \leqslant$$

$$\frac{1}{n}\sum_{i=1}^{n}\mathbb{E}_{y\sim\pi_\star^{\bar{f},\beta}(\cdot|\bar{x}_i)}\left[\bar{f}(\bar{x}_i, y) + \beta\log\left(\pi_\star^{\bar{f},\beta}(y|\bar{x}_i)/\pi_{\mathrm{ref}}(y|\bar{x}_i)\right)\right]. \qquad (42)$$

Since $-c_{\max} \leqslant \mathbb{E}_{y\sim\pi(\cdot|x)}\left[\bar{f}(x,y) + \beta\log\left(\pi(y|x)/\pi_{\mathrm{ref}}(y|\bar{x}_i)\right)\right] \leqslant c_{\max}$ for all $x \in \mathcal{X}$ and all $\pi \in \Pi(\bar{c})$ (see Equation 37) Hoeffding inequality implies that,

$$\left|\left(\frac{1}{n}\sum_{i=1}^{n}J_\beta(\pi, \bar{f}, \bar{x}_i)\right) - \mathbb{E}_{x\sim\rho, y\sim\pi(y|x)}\left[J_\beta(\pi, \bar{f}, x)\right]\right| \leqslant 2c_{\max}\sqrt{\frac{\log(|\Pi|/\delta)}{n}} \qquad (43)$$

holds with probability at least $1 - \delta/4$ for all $\pi \in \Pi(\bar{c})$ simultaneously where $J_\beta(\pi, f, x) = \mathbb{E}_{y\sim\pi(\cdot|x)}\left[f(x,y) + \beta\log\left(\pi(y|x)/\pi_{\mathrm{ref}}(y|x)\right)\right]$.

Combining equations 42 and 43 we conclude that with probability at least $1 - \delta/4$,

$$\begin{aligned}
\mathbb{E}_{x\sim\rho, y\sim\pi(y|x)}\left[J_\beta(\hat{\pi}_{\mathrm{RL}}, \bar{f}, x)\right] &\leqslant \mathbb{E}_{x\sim\rho, y\sim\pi_\star^{\bar{f},\beta}(y|x)}\left[J_\beta(\pi_\star^{\bar{f},\beta}, \bar{f}, x)\right] + 2c_{\max}\sqrt{\frac{\log(2|\Pi|/\delta)}{n}} \\
&\overset{(i)}{\leqslant} \mathbb{E}_{x\sim\rho, y\sim\pi_\star^{c,\beta}(y|x)}\left[J_\beta(\pi_\star^{c,\beta}, \bar{f}, x)\right] + 2c_{\max}\sqrt{\frac{\log(2|\Pi|/\delta)}{n}}. \qquad (44)
\end{aligned}$$

where inequality $(i)$ holds because $\pi_\star^{\bar{f},\beta}$ is achieves a better regularized objective value for cost $\bar{f}$ than $\pi_\star^{c,\beta}$.

Moreover, pessimism implies that when $\mathcal{E}$ holds,

$$\mathbb{E}_{x\sim\rho,y\sim\hat{\pi}_{\text{RL}}(\cdot|x)}\left[c(x,y)+\beta\log(\hat{\pi}_{\text{RL}}(y|x)/\pi_{\text{ref}}(y|x))\right] \leqslant$$
$$\mathbb{E}_{\rho\sim x,y\sim\hat{\pi}_{\text{RL}}(\cdot|x)}\left[\bar{f}(x,y)+\beta\log(\hat{\pi}_{\text{RL}}(y|x)/\pi_{\text{ref}}(y|x))\right].$$

Combining these results we conclude that,

$$\mathbb{E}_{x\sim\rho}\left[J_\beta(\hat{\pi}_{\text{RL}},c,x)\right] \leqslant \mathbb{E}_{x\sim\rho}\left[J_\beta(\pi_\star^{c,\beta},\bar{f},x)\right] + 2c_{\max}\sqrt{\frac{\log(2|\Pi|/\delta)}{n}}. \tag{45}$$

The final step is to upper bound $\mathbb{E}_{x\sim\rho,y\sim\pi_\star^{c,\beta}(\cdot|x)}\left[\bar{f}(x,y)\right]$ in terms of $\mathbb{E}_{x\sim\rho,y\sim\pi_\star^{c,\beta}(\cdot|x)}\left[c(x,y)\right]$.

To do this we will again use $\mathcal{E}$ and Cauchy Schwartz's inequality, in the next few lines we'll use the shorthand $\mathbb{E}_\pi[\cdot]$ to denote $\mathbb{E}_{x\sim\rho,y\sim\pi(\cdot|x)}[\cdot]$.

$$\left|\mathbb{E}_{\pi_\star^{c,\beta}}\left[\bar{f}(x,y)\right]-\mathbb{E}_{\pi_\star^{c,\beta}}\left[c(x,y)\right]\right| = \left|\mathbb{E}_{\pi_{\text{ref}}}\left[\frac{\pi_\star^{c,\beta}(y|x)}{\pi_{\text{ref}}(y|x)}\left(\bar{f}(x,y)-c(x,y)\right)\right]\right|$$

$$\leqslant \sqrt{\mathbb{E}_{\pi_{\text{ref}}}\left[\left(\bar{f}(x,y)-c(x,y)\right)^2\right]} \cdot \underbrace{\sqrt{\mathbb{E}_{\pi_{\text{ref}}}\left[\left(\frac{\pi_\star^{c,\beta}(y|x)}{\pi_{\text{ref}}(y|x)}\right)^2\right]}}_{\bar{C}_\star}$$

Finally we can bound the term $\mathbb{E}_{\pi_{\text{ref}}}\left[\left(\bar{f}(x,y)-c(x,y)\right)^2\right]$. Lemma 12 implies,

$$\mathbb{E}_{\pi_{\text{ref}}}\left[\left(\bar{f}(x,y)-c(x,y)\right)^2\right] \leqslant \frac{136b\cdot c_{\max}^2\log\left(\frac{|\mathcal{F}|}{\delta}\right)\cdot\log(n)\cdot d_{\text{eluder}}(\mathcal{F},c_{\max}/n)}{n}$$

with probability $1-3\delta/4$. And therefore,

$$\mathbb{E}_{x\sim\rho}\left[J_\beta(\hat{\pi}_{\text{RL}},c,x)\right] \leqslant \mathbb{E}_{x\sim\rho}\left[J_\beta(\pi_\star^{c,\beta},c,x)\right] + 2c_{\max}\sqrt{\frac{\log(2|\Pi|/\delta)}{n}} +$$

$$12c_{\max}\bar{C}_\star\sqrt{\frac{b\log\left(\frac{|\mathcal{F}|}{\delta}\right)\cdot\log(n)\cdot d_{\text{eluder}}(\mathcal{F},c_{\max}/n)}{n}}$$

with probability at least $1-\delta$.

$\square$

## A.6 Pure Online Learning Theory

In this section we introduce and analyze the regret of the Epoch Supervised Learning Algorithm (see Algorithm 2). This procedure works in epochs, in each epoch the algorithm observes prompts from $\rho$, and produces responses $y$ from a fixed edits distribution. The algorithm also collects edit data in the form of a samples $y'\sim q(\cdot|x,y)$. During epoch $e$ the algorithm uses policy $\pi_e$ to produce responses and collect $m_e$ datapoints of the form $\mathcal{D}_e=\{x_{e,n},y_{e,n},y'_{e,n}\}_{n=1}^{m_e}$ where $y'_{e,n}\sim q(\cdot|x_{n,e},y_{n,e})$.

The epoch supervised learning algorithm's policy is updated by fitting a policy matching the edited responses in $\mathcal{D}_e$,

$$\pi_{e+1}=\arg\max_{\pi\in\Pi}\sum_{(x,y')\in\mathcal{D}_e}\log\pi(y'\mid x)$$

This way the policy $\pi_{e+1}$ is an approximation to $q \circ \pi_e$. We consider a setting where Algorithm 2 interacts in $t = 1, \cdots, T$ rounds where in each time step the algorithm is given a prompt $x_t$ and generates a response $y_t$. We measure the performance of 2 through its regularized regret $\texttt{Reg}_T$ as defined in Equation 2. Our main result is the following theorem where we show Algorithm 2 satisfies a $\mathcal{O}(\sqrt{T})$ regret bound.

---

**Algorithm 2** Epoch Supervised Learning

---

1: Start arbitrary initial policy $\pi_1 = \pi_{\text{ref}}$.
2: **for** Epoch $e = 1, 2, \cdots, \lfloor e(T) \rfloor$ **do**
3:     $\mathcal{D}_e = \varnothing$
4:     **for** $n = 1, 2, \cdots, m_e$ **do**
5:         Given context $x_{e,n}$
6:         Agent generates $y_{e,n} \sim \pi_e$
7:         User edits it to $y'_{e,n}$
8:         Agent collects edit distance of $\Delta_{\text{edit}}(y_{e,n}, y'_{e,n})$
9:         $\mathcal{D}_e \leftarrow \mathcal{D} \cup \left\{ (x_{e,n}, y'_{e,n}) \right\}$
10:    Update the policy:
$$\pi_{e+1} = \arg\max_{\pi \in \Pi} \sum_{(x,y') \in \mathcal{D}_e} \log \pi(y' \mid x)$$

---

**Theorem 8.** *When Assumption 1 and Assumption 4 hold then Algorithm 2 with inputs $m_e = \frac{2 \ln\left(\frac{|\Pi|}{\delta_e}\right)}{(1 - \gamma_{min})^{2e}}$, $\delta_e = \delta/2e^2$ and $e(T)$ be the first index such that $\sum_{e=1}^{e(T)-1} m_e < T$ and $\sum_{e=1}^{e(T)} m_e \geqslant T$ satisfies the regret bound,*

$$\texttt{Reg}_T \leqslant \mathcal{O}\left( \frac{c_{max} + V_{max}}{(1 - \gamma_{min})^2} \cdot \left( \log_{1/(1-\gamma_{min})} \left( T \ln \left( \frac{T|\Pi|}{\delta} \right) \right) \right)^{3/2} (T) \sqrt{T \ln \left( \frac{T|\Pi|}{\delta} \right)} \right).$$

*with probability at least $1 - \delta$.*

Algorithm 2 works over $e(T)$ epochs. In each epoch the algorithm interacts with the "editor" for $m_e$ steps.

At the start of epoch $e$ we assume access to a response generator $\pi_e(y|x)$. This induces a population version of the target distribution $p_{e+1}(x, y, y')$ defined as $p_{e+1}(x, y, y') = \mathbb{P}_{\mathcal{X}}(x)\pi_e(y \mid x)q(y' \mid x, y)$. This induces the following ideal updated generator $\bar{\pi}_{e+1}(y|x)$ defined as,

$$\bar{\pi}_{e+1}(y|x) = \frac{\sum_{\tilde{y}} p_{e+1}(x, \tilde{y}, y)}{\mathbb{P}_{\mathcal{X}}(x)}$$

Since we admit $m_e$ samples during epoch $e$, the MLE guarantee ensures that the computed generator $\pi_{e+1}$ satisfies,

$$\mathbb{E}_{x \sim \mathbb{P}_{\mathcal{X}}} \left[ \left\| \pi_{e+1}(Y' \mid x) - \bar{\pi}_{e+1}(Y' \mid x) \right\|_{\text{TV}} \right] \leqslant \sqrt{\frac{2}{m_e} \ln \left( \frac{|\Pi|}{\delta_e} \right)} \tag{46}$$

with probability at least $1 - \delta_e$. Lemma 1 implies

$$\mathbb{E}_{x \sim \mathbb{P}_{\mathcal{X}}} \left[ \left\| \bar{\pi}_{e+1}(Y' \mid x) - \pi_{\star}^{\beta}(Y' \mid x) \right\|_{\text{TV}} \right] \leqslant (1 - \gamma_{min}) \cdot \mathbb{E}_{x \sim \mathbb{P}_{\mathcal{X}}} \left[ \left\| \pi_e(Y' \mid x) - \pi_{\star}^{\beta}(Y' \mid x) \right\|_{\text{TV}} \right] \tag{47}$$

therefore, combining Equations 46 and 47 we obtain,

$$\mathbb{E}_{x \sim \mathbb{P}_{\mathcal{X}}} \left[ \left\| \pi_{e+1}(Y' \mid x) - \pi_{\star}^{\beta}(Y' \mid x) \right\|_{\text{TV}} \right] \leqslant (1 - \gamma_{min}) \cdot \mathbb{E}_{x \sim \mathbb{P}_{\mathcal{X}}} \left[ \left\| \pi_e(Y' \mid x) - \pi_{\star}^{\beta}(Y' \mid x) \right\|_{\text{TV}} \right] + \sqrt{\frac{2}{m_e} \ln \left( \frac{|\Pi|}{\delta_e} \right)}$$

we will now unroll this sum.

Let $\delta_e = \mathbb{E}_{x \sim \mathbb{P}_{\mathcal{X}}} \left[ \left\| \pi_e(Y' \mid x) - \pi_\star^\beta(Y' \mid x) \right\|_{\mathrm{TV}} \right]$ and $\xi_e = \sqrt{\frac{2}{m_e} \ln\left( \frac{|\Pi|}{\delta_e} \right)}$.

Then, unrolling the recursion we obtain,

$$\delta_{e+1} \leqslant (1 - \gamma_{\min})\delta_e + \xi_e \leqslant \eta\left( (1 - \gamma_{\min})\delta_{e-1} + \xi_{e-1} \right) + \xi_e = (1 - \gamma_{\min})^2 \delta_{e-1} + (1 - \gamma_{\min})\xi_{e-1} + \xi_e \leqslant \cdots$$

so that $\delta_{e+1} \leqslant \sum_{i=1}^{e}(1 - \gamma_{\min})^{e-i}\xi_i + (1 - \gamma_{\min})^e \delta_1$. Therefore,

$$\delta_e = \sum_{i=1}^{e-1}(1 - \gamma_{\min})^{e-i-1}\xi_i + (1 - \gamma_{\min})^{e-1}\delta_1.$$

Lemma 5 allows us to bound the regret of epoch $e$ by,

$$\mathtt{Reg}_{e,T} \leqslant 2(c_{\max} + V_{\max}) \cdot m_e \cdot \left( \sum_{i=1}^{e-1}(1 - \gamma_{\min})^{e-i-1}\xi_i + (1 - \gamma_{\min})^{e-1}\delta_1 \right)$$

Set $m_e = \frac{2 \ln\left( \frac{|\Pi|}{\delta_e} \right)}{(1 - \gamma_{\min})^{2e}}$ so that $\xi_e = (1 - \gamma_{\min})^e$. Then,

$$\mathtt{Reg}_{e,T} \leqslant 2(c_{\max} + V_{\max}) \cdot m_e \cdot \left( \sum_{i=1}^{e-1}\eta^{e-i-1}\xi_i + (1 - \gamma_{\min})^{e-1}\delta_1 \right)$$

$$= 4(c_{\max} + V_{\max}) \cdot \frac{\ln\left( \frac{|\Pi|}{\delta_e} \right)}{(1 - \gamma_{\min})^{2e}} \cdot (1 - \gamma_{\min})^{e-1} \cdot e$$

$$= 4(c_{\max} + V_{\max})e \cdot \frac{\ln\left( \frac{|\Pi|}{\delta_e} \right)}{(1 - \gamma_{\min})^{e+1}}$$

$$= \frac{2(c_{\max} + V_{\max})e}{1 - \gamma_{\min}}\sqrt{2 \ln\left( \frac{|\Pi|}{\delta_e} \right) m_e}$$

where the last equality follows because $m_e = \frac{2 \ln\left( \frac{|\Pi|}{\delta_e} \right)}{(1 - \gamma_{\min})^{2e}}$.

Setting $\delta_e = \delta/2e^2$ we conclude that

$$\mathtt{Reg}_{e,T} \leqslant \frac{2(c_{\max} + V_{\max})e}{1 - \gamma_{\min}}\sqrt{2 \ln\left( \frac{|\Pi|}{\delta_e} \right) m_e}$$

for all $e \in \mathbb{N}$ simultaneously with probability at least $1 - \delta$.

To prove a regret bound up to time $T$ let $e(T)$ be the necessary epoch index such that $\sum_{e=1}^{e(T)-1} m_e < T \leqslant \sum_{e=1}^{e(T)} m_e$. Since the size of $m_e$ is increasing it follows that,

$$T \leqslant \sum_{e=1}^{e(T)} m_e \leqslant \frac{T}{\eta^2} \cdot \frac{\ln\left( \frac{|\Pi|}{\delta_{e(T)+1}} \right)}{\ln\left( \frac{|\Pi|}{\delta_{e(T)}} \right)} \leqslant \frac{2T}{(1 - \gamma_{\min})^2}.$$

the epoch parameter $e(T)$ does not need to be greater than $e(T) \leqslant \left\lceil 2 \log_{1/(1-\gamma_{\min})}\left( T/2 \ln\left( \frac{2T^2|\Pi|}{\delta} \right) \right) \right\rceil$ since at this point,

$$T \leqslant \sum_{e=1}^{\left\lceil 2 \log_{1/(1-\gamma_{\min})}\left( T/2 \ln\left( \frac{2T^2|\Pi|}{\delta} \right) \right) \right\rceil} m_e.$$

And therefore

$$\texttt{Reg}_T \leqslant \frac{2(c_{\max} + V_{\max})}{1 - \gamma_{\min}} \sum_{e=1}^{e(T)} e\sqrt{2\ln\left(\frac{2T^2|\Pi|}{\delta}\right) m_e}$$

$$\leqslant \mathcal{O}\left(\frac{c_{\max} + V_{\max}}{1 - \gamma_{\min}} e(T)\sqrt{e(T)\ln\left(\frac{T|\Pi|}{\delta}\right) \sum_{e=1}^{e(T)} m_e}\right)$$

$$\leqslant \mathcal{O}\left(\frac{c_{\max} + V_{\max}}{1 - \gamma_{\min}} e(T)\sqrt{e(T)\ln\left(\frac{T|\Pi|}{\delta}\right) \frac{T}{\eta^2}}\right)$$

$$= \mathcal{O}\left(\frac{c_{\max} + V_{\max}}{(1 - \gamma_{\min})^2} e^{3/2}(T)\sqrt{T\ln\left(\frac{T|\Pi|}{\delta}\right)}\right)$$

with probability at least $1 - \delta$. Finally since $e(T) \leqslant \left\lceil 2\log_{1/(1-\gamma_{\min})}\left(T/2\ln\left(\frac{2T^2|\Pi|}{\delta}\right)\right)\right\rceil$ the result follows.

## A.7 Support Results

We state useful supporting results below that allow us to prove our main results.

**Lemma 13** (Sigmoid Lower Bound). *For any $x, y \in \mathbb{R}$ we have:*

$$|\sigma(x) - \sigma(y)| \geqslant \min\left\{\sigma'(x), \sigma'(y)\right\} \cdot |x - y|$$

*Proof.* We have $\sigma(t) = \frac{1}{1+e^{-t}}$ for any $t \in \mathbb{R}$. This gives $\sigma'(t) = \frac{e^{-t}}{(1+e^{-t})^2} = e^{-t}\sigma(t)^2$. Observe that $\sigma'(t) \geqslant 0$ for all $t \in \mathbb{R}$. We also have

$$\begin{aligned}
\sigma''(t) &= -e^{-t}\sigma(t)^2 + 2e^{-t}\sigma(t)\sigma'(t) \\
&= -\sigma'(t) + 2e^{-t}\sigma(t)\sigma'(t) \\
&= \sigma'(t)\left(2e^{-t}\sigma(t) - 1\right) \\
&= \sigma'(t)\frac{(1 - e^t)}{1 + e^t}.
\end{aligned}$$

Therefore, $\sigma''(t) \geqslant 0$ (convexity condition) if and only if $1 \geqslant e^t$. Therefore, $\sigma$ is concave on $[0, \infty)$ and convex on $(-\infty, 0]$.

Fix $x, y \in \mathbb{R}$ and assume $x < y$ without loss of generality. Note that the result is trivially true for $x = y$. As $\sigma$ is differentiable over $\mathbb{R}$, we have from mean value theorem $\frac{\sigma(y) - \sigma(x)}{y - x} = \sigma'(u)$ for some $u \in [x, y]$. As $\sigma'(t) \geqslant 0$ for all $t \in \mathbb{R}$, we have $|\sigma(y) - \sigma(x)| = \sigma'(t)|y - x|$.

We now consider 3 cases.

1. If $x \leqslant y \leqslant 0$ in which case we are in the convex regime and $\sigma'(u) \geqslant \sigma'(x)$

2. If $y \geqslant x \geqslant 0$ in which case we are in the concave regime and $\sigma'(u) \leqslant \sigma'(y)$.

3. If $x \leqslant 0 \leqslant y$, then if $u \leqslant 0$, then we have $\sigma'(u) \geqslant \sigma'(x)$. And if $u \geqslant 0$, then we have $\sigma'(u) \geqslant \sigma'(y)$.

Putting it together, we always get $\sigma'(t) \geqslant \min\left\{\sigma'(y), \sigma'(x)\right\}$ giving us the desired result. $\square$

**Lemma 14** (Sigmoid Derivative Bound). *If $|x| \leqslant c$ then $\sigma'(x) \geqslant \sigma'(c)$.*

*Proof.* If $x \geqslant 0$, then $-c \leqslant 0 \leqslant x \leqslant c$. As $x$ is in the concave regime of $\sigma$ we have $\sigma'(x) \geqslant \sigma'(c)$. Alternatively, if $x \leqslant 0$, we have $-c \leqslant x \leqslant 0 \leqslant c$, then $x$ is in the convex regime of $\sigma$ and we have $\sigma'(x) \geqslant \sigma'(-c)$. As, $\sigma'(-c) = \frac{e^c}{(1+e^c)^2} = \frac{e^{-c}}{(1+e^{-c})^2} = \sigma'(c)$, we get $\sigma'(x) \geqslant \sigma'(c)$ in all cases. $\square$

**Lemma 15** (Log Lower Bound). *For any $t > 0$, we have $|\log(t)| \geqslant \frac{|t-1|}{t+1}$.*

*Proof.* We have $\log(u) \geqslant \frac{u-1}{u+1}$ for $u \geqslant 1$. To prove this let $g(u) = \log(u) - \frac{(u-1)}{u+1}$ then $g(1) = 0$ and $g'(u) = \frac{1}{u} - \frac{2}{(u+1)^2} = \frac{u^2+1}{u(u+1)^2}$. This means $g'(u) \geqslant 0$ for $u \geqslant 1$, and so $g(u) \geqslant 0$ for $u \geqslant 1$.

If $t \geqslant 1$, then $|\log(t)| = \log(t) = \frac{t-1}{t+1} = \frac{|t-1|}{t+1}$. If $t \leqslant 1$, then $|\log(t)| = -\log(t) = \log(1/t) = \frac{1/t-1}{1/t+1} = \frac{|1-t|}{t+1}$. This completes the proof. $\qquad\square$

## B   Related Work

**Theory**   Although there is no theory prior work on edits that we are aware of, there is a rich literature studying theoretical guarantees for RLHF objectives from preference feedback. The works of [61] and [53] pioneered the study of reinforcement learning from human feedback via offline preference data, while works such as [46] and [55]. These preliminary results were mostly concerned with tabular scenarios. More recently, many works have been dedicated to the study of function approximation regimes for learning policies from preference feedback in RLHF scenarios, such as the introduction of the XPO algorithm for exploratory RLHF [54] and [58] for provable guarantees for RLHF from offline data as well as unifying frameworks that bridge online and offline preference feedback [13]. Beyond policy optimization, theoretical insights have also been developed for auxiliary phenomena in RLHF. For example, [26] characterized the self-improvement dynamics of iteratively refined policies, and [56] analyzed the statistical efficiency of Best-of-N mechanisms in preference alignment.

**Modeling edits.**   Prior work on text edits has explored various modeling approaches, including generative diffusion models with edit-based corruption and reconstruction [41], latent vector representations of edits [24, 35], and modeling modular operations in the editing process [50, 34, 3, 59]. While these approaches effectively model specific aspects of text editing, our work unifies multiple forms of user feedback — preferences, supervised labels, and cost — within a single framework. These types of feedback, typically studied in isolation, are combined and analyzed both theoretically and empirically in this work.

**Using edits.**   Research on text revision often focuses on task-specific improvements, such as enhancing model factuality [12, 33, 7], refining academic writings [36, 30, 18], and enabling style transfer [42]. Similarly, work on code edits has primarily aimed at automating program repair [57, 16, 41, 60, 49, 9]. In contrast, we focus on user-driven text edits that reflect expectations and preferences in practical use cases. Building on the setup proposed in prior work [?, 4], we systematically investigate the algorithms and learning challenges associated with leveraging such edits.

## C   Additional Experimental Details

We provide details of our experimental setup here. We used the Prelude framework of Gao et al., [21] which is available at `https://github.com/gao-g/prelude` and has MIT License. We use the following models from HuggingFace: Llama 3.1 8b Instruct, Llama 3.3 70b Instruct and Qwen3-32B.

**Grid Search.**   We provide hyperparameter values of `SFT` in Table 5 and for `DPO` and `EarlyEnsemble` in Table 6. We selected the best hyperparameters by computing log-loss of user-edits given context on a held-out validation set with 200 examples.

| Hyperparameter | Values |
|---|---|
| Pretrained Model | meta-llama/Llama-3.1-8B-Instruct |
| Precision | Bfloat16 |
| Optimizer | AdamW, betas=(0.9, 0.95) |
| Learning Rate | {1.0e-5, 1.0e-6, 1.0e-7} |
| Epoch | {1ep, 2ep} |

Table 5: Hyperparameter search ranges for SFT.

| Hyperparameter | Values |
|---|---|
| Pretrained Model | `meta-llama/Llama-3.1-8B-Instruct` |
| Precision | `Bfloat16` |
| Optimizer | `AdamW, betas=(0.9, 0.95)` |
| DPO beta | $\{0.1, 0.5\}$ |
| $\lambda$ in `EarlyEnsemble` (Equation 7) | $\{0, 0.5, 0.75, 1.0\}$ |
| Learning Rate | $\{1.0e\text{-}6, 3.0e\text{-}7, 7.0e\text{-}7\}$ |
| Epoch | $\{2ep\}$ |

Table 6: Hyperparameter search ranges for `DPO` ($\lambda = 0$) and `EarlyEnsemble` ($\lambda > 0$).

For `LateEnsemble` (Algorithm 1) we set $\alpha = 150$ based on the approximate average edit cost of the base model. We did not tune $\alpha$.

**Inference.** We do greedy decoding and allow the agent to generate at most 1000 tokens.

**Compute and Time.** We ran our experiments on a cluster with H100s and H200s. All experiments took a few hours to train. The inference took 10min for each seed for `SFT`, `DPO` and `EarlyEnsemble` for each seed since we can evaluate each datapoint in parallel, while it took 30min for each seed for `LateEnsemble` (Algorithm 1) since we cannot evaluate datapoints in parallel for this.

**Datasets.** We do not release any new dataset with this paper. However, we used source articles from four datasets provided below. The link to the dataset contains other details such as size of the dataset, samples, and license. We urge readers to check the current status of these datasets in context of this work, and recommend them to avoid using any dataset that has been deprecated.

| Domain | Link to the Dataset |
|---|---|
| PG19[] | `https://github.com/google-deepmind/pg19` |
| Arxiv | `https://huggingface.co/datasets/CShorten/ML-ArXiv-Papers` |
| BillSum | `https://huggingface.co/datasets/FiscalNote/billsum` |
| Elsevier OA CC-By [] | `https://huggingface.co/datasets/orieg/elsevier-oa-cc-by` |

Table 7: List of publicly available datasets that we use for training and evaluation.

We provide additional experimental results below.

**Test-time plots.** Figure 3 shows the cumulative edit cost at test time for the email writing setting in Table 1. We also show cumulative user-edits at inference time for the transfer learning setting in Table 2 in Figure 4. The trend is similar to the trend in Figure 3 with `LateEnsemble` mostly converging to the performance of the best method. We can see that in all domain except email writing with a weak user, the performance of `LateEnsemble` gradually drifts towards the best performing method as we would expect. The anomalous setting of email writing with weak user may need more test rounds before `LateEnsemble` converges to the best method.

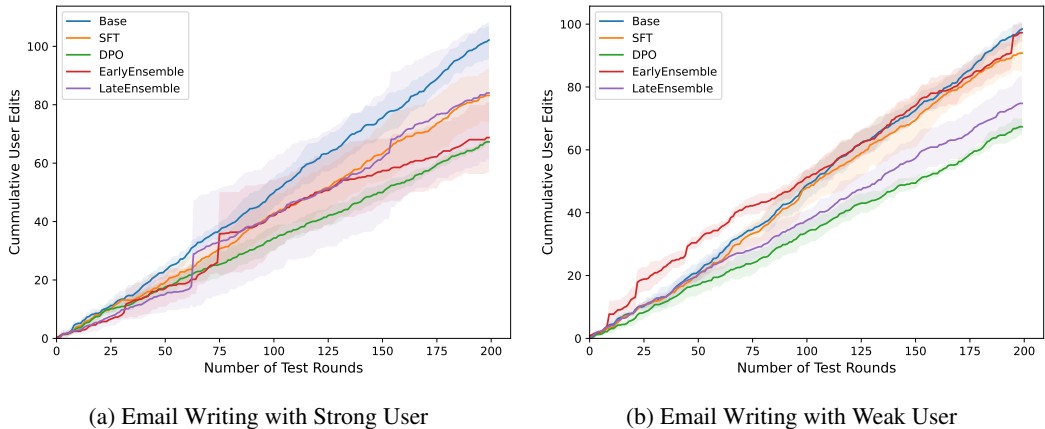

(a) Email Writing with Strong User

(b) Email Writing with Weak User

Figure 3: Cumulative User Edits at test time corresponding to Table 1 for the different setup. For each round, we show mean and standard deviation across 3 seeds.

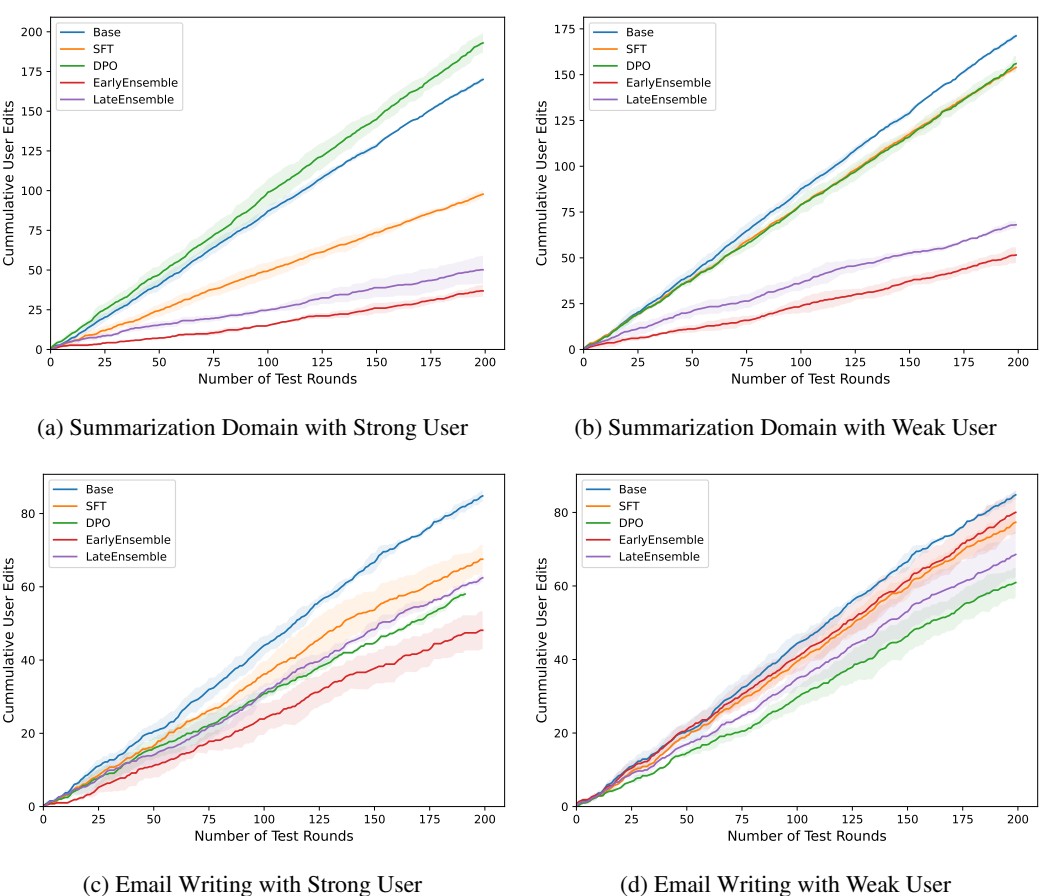

(a) Summarization Domain with Strong User

(b) Summarization Domain with Weak User

(c) Email Writing with Strong User

(d) Email Writing with Weak User

Figure 4: Cumulative User Edits at test time corresponding to Table 2 for the different setup. For each round, we show mean and standard deviation across 3 seeds.

