# OpenReview forum: "Principled Fine-tuning of LLMs from User-Edits: A Medley of Preference, Supervision, and Reward"
_NeurIPS.cc/2025/Conference — NeurIPS 2025 poster_

### Official Review · Reviewer_k1w2 · 2025-06-30

**Clarity:** 3
**Significance:** 2
**Originality:** 2
**Rating:** 5
**Confidence:** 3

**Summary:**

This paper pioneers a principled framework for fine-tuning LLMs using user-edits (e.g., from writing assistants or coding agents). Derive bounds are first derived for learning algorithms that learn from each of these feedback types. A simple ensembling procedure is than proposed to jointly learn from these feedback types.

Three feedback types, supervision, preferences, and rewards are integrated into a single theoretical and empirical paradigm. The authors derive sample complexity bounds for each approach, revealing trade-offs influenced by user behavior.

To balance these, they propose early and late ensemble methods. Experiments on email/summarization tasks show LateEnsemble’s robustness to user distribution shifts, reducing edit costs by a large margin.

Strengths include theoretical grounding and practical algorithms; limitations involve future usage of pessimistic RL and simulated user validation.

**Questions:**

1. Could prompt-based adaptation (e.g., in-context learning) complement fine-tuning for user personalization? and How? prefer to learn more details.
2. any discussion on handling users' edit data privacy topics?
3. can you conclude your contributions at the introduction section?

**Ethical Concerns:**

["NO or VERY MINOR ethics concerns only"]

**Final Justification:**

I upgrade my score by reading the authors' rebuttal contents and comments from other reviewers and appended results/conents from the authors. This paper shall be interested by a relatively wide range of researchers of this field.

**Limitations:**

yes

**Paper Formatting Concerns:**

good format

**Quality:**

2

**Strengths And Weaknesses:**

Strong:
1. the motivation of using user edit feedbacks is relatively interesting and helpful in real-world llm usages. the unifying of perference, supervision and reward learning is a fine-grained method for learning from the users' edition datasets.
2. three complexity bounds for the three learning approaches are derived under relatively clear assumptions. these mathmatical analysis shall help related researchers to obtain more insights of better leveraging sft/rl algorithms for this direction.
3. relatively strong experiments on email/summarization tasks with simulated users and showed that the effectiveness of their proposed methods.

Weak:
1. prefer to learn more connections between the theoretical bounds with more real-world validities, currently the discussion is short.
2. human evaluation is limitted, and more real-world human usage is preferred.
3. scalability requires more evidences such as larger datasets. More discussion or experiments on this direction is appreciated.

---

> ### Author Rebuttal · Authors · 2025-07-31
>
> We thank the reviewer for useful feedback. We address their questions below:
>
> ## 1. Prompt-Based Adaptation
>
> One can easily combine prompt-engineering approaches with fine-tuning approaches. E.g., one can fine-tune an LLM and still prompt it with previously seen relevant in-context learning examples (a popular prompt adaptation approach). While prompt-based adaptation is orthogonal to our main focus on fine-tuning, we will add a short discussion on results with a prompt-adaptation approach. We will try to share preliminary results during the author-reviewer discussion period itself.
>
> $ $
> ## 2. Users' edit data privacy topics
>
> We included a discussion on how to best protect the user’s privacy when handling their user-edit data in the NeurIPS checklist. Please see our response to the broader impact on lines 756-764. If accepted, we will use the extra page to move this discussion to the main paper.
>
> $ $
>
> ## 3. Scalability and Human Studies
>
> In this paper, we focus on laying the algorithmic foundations of learning from user-edits. Expanding our study to a larger scale and doing human studies are natural follow-up work. These studies can refer to our paper to decide what algorithms to run. In terms of scalability, both individual algorithms and their ensembling approach, including Algorithm 1, are easy to run with any standard LLM fine-tuning library. Algorithms such as SFT and DPO are routinely used to train LLMs with big datasets.
>
> One advantage of using simulated LLMs in our study is that they provide reproducible and quick studies. This is especially useful for algorithm development, which is our main focus. The use of simulated users has increasingly become common due to improvements in LLMs [1, 2, 3, 4]. Further, doing a real human study by collecting enough training data and having to do repeated human evaluations can be prohibitively expensive.
>
> However, any industry lab that deploys a writing or coding assistant can easily test our algorithms with a real human study. They can do this by collecting user edits and using them to fine-tune LLMs. They can then do A/B testing to compute total user edits on generations of the trained model (assuming they have all the required permissions to store and train on user edits). We feel our research is particularly suited to make a practical impact in these settings.
>
> References:
> 1. Reliable LLM-based User Simulator for Task-Oriented Dialogue Systems, Sekulić et al. 2024
> 2. Quantifying the Persona Effect in LLM Simulations, Hu and Collier (ACL 2024)
> 3. Simulating User Agents for Embodied Conversational-AI, Philipov et al. 2024
> 4. USimAgent: Large Language Models for Simulating Search Users, Zhang et al., SIGIR 2024
>
>
> $ $
>
> ## 4. Theory and Real World Validity
>
> In section 4.2, we provide a variety of insights on how our theoretical results relate to practice (lines 248 to 255) where we discuss when SFT, DPO or RL are preferable depending on different user edit ``parameters" such as $\gamma_{\min}$ or the preference coverage $C_{\mathrm{PREF}}$ parameter. When deployed, these parameters have interpretable descriptions that characterize how thorough user edits are when generating a response ($\gamma_{\min}$),  or the richness of the edits training data ($C_{\mathrm{PREF}}$). If accepted, we will use the extra page to make some of these connections clearer.
>
> $ $
>
> ## 5. Contributions in Intro
>
> We will revise the introduction to explicitly list our contributions (theoretical bounds, unified framework, and ensemble strategies).

---

### Official Review · Reviewer_kYmZ · 2025-07-01

**Clarity:** 2
**Significance:** 3
**Originality:** 3
**Rating:** 5
**Confidence:** 2

**Summary:**

This paper studies the problem of post-training LLMs to learn from user edits. The proposed learning setup consists of two phases: the offline phase where a sufficiently large dataset of user edits can be used to fine-tune the LLM, and the online phase where the LLM interacts with users in deployment over a few episodes and can potentially be adapted. The paper considers three different approaches for post-training during the offline phase: supervised fine-tuning (SFT), direct preference optimization (DPO), and reinforcement learning with a learned cost function (RL). The paper provides a theoretical analysis of the number of samples required for bounded regret guarantees in the three cases under certain assumptions on the policy class, and the user edits distribution -- further stating the trade-offs between the three methods. An early ensembling of the approaches is also considered. A late ensembling approach is proposed for the online phase: a bandit algorithm is run to select between the the different learned policies from the offline phase based on the cost of user edits.  Finally, the paper presents empirical results on two tasks (summarization and email writing) with strong / weak users simulated using LLMs -- demonstrating the theoretical trade-offs in practice.

**Questions:**

1. Could the authors provide some insight into why DPO is significantly worse than the base method on summarization with a different strong user (Table 2)?
2. The empirical results in Section 5 use Llama-3.3-70B-instruct as the user and Llama-3.1-8B-instruct as the agent. I would expect these models to be very similar since they belong to the same model family (similar pre-training data, post-training approaches possibly leading to very similar generation distributions). Would the results in Table 1 be different if the user and agent are from different model families? I would encourage the authors to present some preliminary results on this if possible.

Other comments:
1. Line 59: "one can use induce a preference data" -> this doesn't sound right
2. Line 70: "approach perform better" -> "approach performs better"
3. Line 109: line 6 -> line 4 (incorrect line number from protocol 1)
4. Line 267: "The use an LLM" -> "The use of an LLM"
5. Line 291: "We perform generate" -> "We generate"
6. Line 303: "Equation 6 give challenges" -> "Equation 6 given challenges"
7. Line 325: "this affect" -> "this effect"
8. Line 337: "these maybe dependent" -> "these may be dependent"

**Ethical Concerns:**

["NO or VERY MINOR ethics concerns only"]

**Final Justification:**

The paper studies a well-motivated and timely problem, adapting LLMs to user preferences using naturally available user edit data.
The theoretical analysis of the different post-training approaches (SFT, DPO, RL with a learned cost function) is original in the context of learning from user edits and the four assumptions on the user edit distribution and policy class appear reasonable.
The empirical evaluation with a strong and weak user provides interesting insights (Section 5) about the trade-offs between the different approaches in terms of average regret with respect to the best policy.

**Limitations:**

yes

**Quality:**

3

**Strengths And Weaknesses:**

**Strengths**
1. The paper studies a well-motivated and timely problem, adapting LLMs to user preferences using naturally available user edit data.
2. The theoretical analysis of the different post-training approaches (SFT, DPO, RL with a learned cost function) is original in the context of learning from user edits and the four assumptions on the user edit distribution and policy class appear reasonable.
3. The empirical evaluation with a strong and weak user provides interesting insights (Section 5) about the trade-offs between the different approaches in terms of average regret with respect to the best policy.

**Weaknesses**

1. Protocol 1 is possibly incomplete: there is a mismatch between the line numbers in protocol 1 and the associated discussion on lines 107-113. For example, line 111 mentions line 9 in protocol 1 while the protocol has 7 lines in total.

---

> ### Author Rebuttal · Authors · 2025-07-31
>
> We thank the reviewer for their helpful feedback. We answer their questions below:
>
> $ $
>
> ## 1. Protocol 1 Line Number Mismatch
>
> **We want to clarify that Protocol 1 is correctly written and complete.** However, there are two typos in the text in the line numbers. On page line 109, the protocol line should be line 4, and on page line 111, the protocol line should be line 7. We appreciate the reviewer for catching this typo, and we will fix these in the revision.
>
> $ $
>
> ## 2. Using different LLMs for teacher and user models
>
> We want to clarify that we experimented with using a different combination of LLMs for user and teacher at test time. In Table 2 in the main paper and in Table 6 in the Appendix, we tested with a setting where the user was either a Qwen or Gemma model while the agent LLM remained Llama 8b. Please also see the discussion on lines 334-342. These results add a new dimension of generalization to our studies by showing the robustness to user LLM. In both cases, the results were generally the same as in Table 1. Notably, our late-ensemble approach performed the best overall.
>
> If, however, the reviewer meant having different LLM families (e.g., Qwen vs Llama) for the user and teacher during __training time__ and then using the same families at test time, then we are also happy to add those results in our revision. This will be an easier setting since, unlike the transfer learning results, the model will be training and testing on the same combination of LLMs.
>
> $ $
>
> ## 3. Why DPO Underperforms the Base Method (Table 2)
>
> We have noticed some issues with DPO, including the well-known observation that DPO-trained models can be very verbose. This verbosity can be especially hurtful in the summarization task, where the goal is to generate a concise summary. We also noticed that DPO can enter repetition loops, where it repeats the same text, something that we fix with an explicit filter (see discussion around lines 306-308). We found that in this particular case of summarization with a strong user and a different LLM, the DPO was generating either really long summaries or really short ones, both of which are sub-optimal. We believe this is due to the instability in DPO, and testing with a different LLM model results in a higher edit distance penalty for these deficiencies.

---

> > ### Comment · Reviewer_kYmZ · 2025-08-05
> >
> > Thank you for the response. I have some follow-up questions that I'd appreciate answers for:
> >
> > 1. Could you please clarify what Protocol 1 is intended to do? I assumed that the Protocol was to fine-tune the agent based on the edits (since the Protocol is titled "Finetuning LLMs from User Edits") and hence there should be some step that uses the edit to update the agent (since lines 110-111:  "the goal of the agent is to minimize the total cost"). The Protocol just returns the total cost, however, without this "minimization" step listed anywhere.
> >
> > 2. I apologize for not specifying which user / agents I was talking about here. I was indeed talking about users / agents at training time which seem to be from the same model family (Llama). I was referring to the setting where the users at training time, at testing time, and the agent are from three different families. So, maybe a Qwen model as a user during training, the Llama 8b model as an agent, and a model from a third family as a user during testing (since that might reflect sufficient diversity). Could the authors comment on this?
> >
> > 3. The discussion on DPO is very interesting. Thank you.

---

> > > ### Author Response · Authors · 2025-08-07
> > > **Clarification**
> > >
> > > Thank you for the response. We provide clarification below. Please let us know if you have any further questions or need more clarification.
> > >
> > > > Could you please clarify what Protocol 1 is intended to do?
> > >
> > > Protocol 1 defines our machine learning problem by showing what are the available inputs (e.g., dataset $\mathcal{D}$), what is available for training, and how does the evaluation work. The goal is to define our problem upfront before discussing any solutions to it. This follows the norm in interactive learning and RL literature of separating problem definition from algorithms. E.g., see Protocol for multi-armed bandit problem on page 8 of [1], contextual bandit Protocol in Algorithm 1 of [2], or learning from edits protocol in [3] (see references below). To clarify, _Protocol 1 does not present a solution to the problem but rather defines our problem_.
> > >
> > > Any algorithm implementing this Protocol 1 needs to implement Line 1 and Line 4 of the protocol. In Line 4 specifically, an algorithm needs to implement how to generate a response $y_t$ given all past information including previous contexts, responses, user edits, and edit distances.
> > >
> > > As an example, Algorithm 1 is an implementation of Protocol 1 where it uses the dataset $\mathcal{D}$ to train a set of policies and then runs a bandit algorithm on them in the testing loop. However, future works may propose newer algorithms for Protocol 1.
> > >
> > > > I apologize for not specifying which user / agents I was talking about here. ... Could the authors comment on this?
> > >
> > > Thank you for clarification. If we understand correctly, what you are suggesting is a setting where we train with a user and agent that are from two different model families, and then test with a user that is from a third model family. We are happy to add this experiment in the revision, though we feel that results in Table 1, Table 2 and Table 6 should be enough to show the generality of our claims.
> > >
> > > $ $
> > >
> > > ### References:
> > >
> > > 1. Introduction to Multi-Armed Bandits, Slivkins et al., 2024, https://arxiv.org/pdf/1904.07272
> > > 2. A Contextual Bandit Bake-off, Bietti et al. 2021 JMLR, https://www.jmlr.org/papers/volume22/18-863/18-863.pdf
> > > 3. Aligning LLM Agents by Learning Latent Preference from User Edits, Gao et al., NeurIPS 2024 https://arxiv.org/pdf/2404.15269

---

> > > > ### Comment · Reviewer_kYmZ · 2025-08-08
> > > >
> > > > Thank you for the response.
> > > >
> > > > If Protocol 1 is intended to define the problem, I would expect the last step (line 7) to say "Update / evaluate agent based on the total cost" (or something similar) as opposed to "Return total cost" (similar to the learning from edits protocol in [3]).

---

> > > > > ### Author Response · Authors · 2025-08-08
> > > > > **Thank You**
> > > > >
> > > > > Thank you for the response.
> > > > >
> > > > > > If Protocol 1 is intended to define the problem, I would expect the last step (line 7) to say "Update / evaluate agent based on the total cost" (or something similar) as opposed to "Return total cost" (similar to the learning from edits protocol in [3]).
> > > > >
> > > > > We have mentioned this in the text (see Line 110-111), _``The goal of the agent is to minimize the total edits performed during these $T$ episodes"_, but based on your feedback, we will also add this clarification in Line 7 of Protocol 1 as well.

---

### Official Review · Reviewer_U2M6 · 2025-07-03

**Clarity:** 4
**Significance:** 2
**Originality:** 4
**Rating:** 5
**Confidence:** 3

**Summary:**

The paper tackles fine‑tuning LLMs from user‑edit logs—situations where a user edits an LLM’s draft. It formalizes how edits simultaneously provide: (i) supervised targets, (ii) pairwise preferences, and (iii) a reward signal proportional to edit distance.

The authors derive sample‑complexity bounds for learning from each signal, reveal their trade‑offs, and introduce two ensembling strategies. The LateEnsembl, UCB bandit that picks among separately fine‑tuned policies, which yields the lowest edit cost on simulated email‑writing and summarization tasks.

**Questions:**

1. Could you clarify how critical Assumption 1 (the contraction with rate $\gamma_{\min}$) and other assumptions are in practice? For instance, if real users sometimes do not fully correct every error (violating the contraction property) or if $\gamma_{\min}$ varies across contexts, how would that affect the learning outcome?

2. What specific obstacles prevented implementing Eq. 6?

**Ethical Concerns:**

["NO or VERY MINOR ethics concerns only"]

**Quality:**

3

**Strengths And Weaknesses:**

1. The paper is strong in theoretical quality. And it  tackles a timely and practically significant problem – leveraging naturally occurring user edits to improve LLMs. Unlike expensive hand-labeled data (human-written gold answers or explicit preference ratings), edit logs are abundant and user-generated. The authors are among the first to provide a unified framework for this scenario, identifying that preferences, demonstrations, and reward signals co-occur in user-edit data.
2. The paper proposes a simple yet clever ensembling strategy to get the best of all worlds. Instead of committing to one feedback type, the approach either (i) blends the training objectives, or (ii) trains separate models for each and then uses an online bandit algorithm (UCB) to choose which policy to deploy at test time.
3. A notable weakness is the lack of real human user data or human evaluation. All experiments rely on a simulated user (an LLM that performs edits based on preset preferences). While this is a reasonable proxy to validate the concepts, it remains uncertain how well the proposed methods would work with actual users. Real user edits might be noisy, inconsistent, or exhibit biases that a simulated LLM user does not. The paper would be stronger if it at least discussed or tested the approach on a small real-world dataset (if available) or provided a plan for such validation.
4. The work identifies three feedback modalities (supervision, preference, reward), but in practice the pure reward-based approach (RL with edit distance) was not fully implemented in the experiments.
5. The theoretical analysis, while insightful, relies on several strong assumptions (explicitly labeled Assumption 1–3 in the paper).

---

> ### Author Rebuttal · Authors · 2025-07-31
>
> We thank the reviewer for their detailed feedback. We address their questions below:
>
> ## 1. Clarification on assumptions and their validity
>
> We first provide some important clarifications regarding our assumptions.
>
> > “How is the learning outcome affected if real users sometimes do not fully correct every error (violating the contraction property)”
>
> We do not assume that users fully correct every error. In fact, this would be a very strong assumption. The contraction lemma states that the user will improve a given policy's response with the rate of improvement determined by $\gamma_\min$. This does not mean the user will fix all issues in one round of edits. Instead, what this implies is that if the user keeps on editing a given response to make it better and better, then they will eventually converge to their optimal policy. This is a much weaker assumption and is similar to our lived experience, where constantly editing one’s work (e.g., for a paper deadline) leads to an eventually acceptable version.
>
> > “How is the learning outcome affected if $\gamma_{\min}$ varies across contexts”
>
> This is a great question. Suppose we have a context $x$ dependent value $\gamma_{\min}(x)$ where Equation 9 holds. The simplest way will be to define $\gamma_\min = \min_{x \in X} \gamma_{\min}(x)$ as the minimum over all contexts. However, we can do even better as follows:
>
> Firstly, in Lemma 1, for any context $x$, we will get:
>
> $||q \circ \pi_{ref}(. | x) - \pi^\beta_\star(. | x) || \le (1 - \gamma_\min(x)) ||\pi_{ref}(.|x) - \pi^\beta_\star(. | x)||$,
>
> Then, in the proof of Theorem 4, we can use Cauchy-Schwarz’s inequality to get:
>
> $E[||q \circ \pi_{ref}(. | x) - \pi^\beta_\star(. \mid x) ||] \le E[(1 - \gamma_\min(x)) ||\pi_{ref}(.|x) - \pi^\beta_\star(.|x) ||]$
>
> $\qquad \qquad\qquad \qquad \qquad \qquad \le \sqrt{E[(1 - \gamma_\min(x))^2]} \sqrt{E[||\pi_{ref}(.|x) - \pi^\beta_\star(.|x) ||^2]} $
>
> where we now only require $\sqrt{E_{x}[(1 - \gamma_\min(x))^2]}$ to be small. This is a _weaker assumption_ as $\gamma_\min(x)$ no longer needs to be close to 1 for all contexts. We will update our proof to use this softer notion of $\gamma_\min$.
>
> More generally, we believe our results are robust to assumptions being violated. The biggest evidence is our experiments, where the assumptions may not necessarily hold as we use complex transformer-based LLMs and perform optimizations that may not necessarily converge to an ERM solution; however, findings from our theory still apply. Finally, assumptions similar to ours are routinely made in RLHF and RL literature.
>
>
>
> ## 2. Challenges of Implementing Equation 6
>
> Unfortunately, computing an ``exact’’ pessimistic $\bar{f}$ estimator for $f$ requires solving a challenging optimization problem (see line 157 for a definition of $\bar{f}$). This is because enforcing the constraint set $\gamma(\mathcal{F}, \delta)$ is impractical to achieve in real settings. There are some heuristics to implement pessimism using an ensemble, but these tend to be brittle in practice and/or have their own challenges. Our theoretical analysis nevertheless provides insights into the potential of this approach. We will add this clarification in the revision.

---

### Decision · Program_Chairs · 2025-09-17

**Decision:**

Accept (poster)

**Comment:**

Summary: This paper introduces a theoretical framework for fine-tuning LLMs from user-edit data by unifying three feedback types (supervision, preference, and reward).

Strengths: All the reviewers agree upon the paper's primary strengths: its novel and well-motivated theoretical framework that unifies multiple feedback types from user edits, and the principled derivation of algorithms with clear trade-offs.

Weaknesses: The primary point of discussion was the reliance on a simulated user environment for experiments. Evaluating on real human data is a valuable direction for future work but does not detract from the current paper's contributions.

Reasons for Decision: The paper is accepted based on the strong and unanimous consensus from the reviewers. All reviewers agree that this paper's theoretical contribution: providing a principled and unified foundation for learning from user edits. And it is a significant and timely advance for a practical problem. While the reliance on simulated users is a limitation, the conceptual framework and algorithmic insights are of high interest to the community and outweigh this concern.

Discussion Process: The authors successfully clarified technical questions regarding the experimental protocol and theoretical assumptions for the reviewers, and directly led one reviewer to upgrade their score, solidifying a unanimous recommendation for acceptance.